# VidText: Towards Comprehensive Evaluation for Video Text Understanding

## Abstract

Visual texts embedded in videos carry rich semantic information, which is crucial for both holistic video understanding and fine-grained reasoning about local human actions. However, existing video understanding benchmarks largely overlook textual information, while OCR-specific benchmarks are constrained to static images, limiting their ability to capture the interaction between text and dynamic visual contexts. To address this gap, we propose VidText, a new benchmark designed for comprehensive and in-depth evaluation of video text understanding. VidText offers the following key features: 1) It covers a wide range of real-world scenarios and supports multilingual content, encompassing diverse settings where video text naturally appears. 2) It introduces a hierarchical evaluation framework with video-level, clip-level, and instance-level tasks, enabling assessment of both global summarization and local retrieval capabilities. 3) The benchmark also introduces a set of paired tasks for perception and reasoning, ranging from visual text perception to cross-modal reasoning between textual and visual information. Extensive experiments on 18 state-of-the-art Large Multimodal Models (LMMs) reveal that current models struggle across most tasks, with significant room for improvement. Further analysis validates the effectiveness of VidText—spanning joint video-text and multimodal reasoning, multi-granularity task structure, and temporal modeling. It also reveals substantial effects from model-intrinsic factors (input resolution, OCR capability) and external factors (auxiliary context and video-text-centric Chain-of-Thought strategies). We hope VidText will fill the current gap in video understanding benchmarks and serve as a foundation for future research on multimodal reasoning with video text in dynamic environments.

## 1 Introduction

Large Multimodal Models (LMMs) (Bai et al., 2025; Chen et al., 2024d; Liu et al., 2023b; Alayrac et al., 2022; Li et al., 2023a) are rapidly emerging as general-purpose solutions for a wide range of vision–language tasks, demonstrating impressive perception and cognitive capabilities across various multimodal benchmarks. Building on this success, there is a growing interest in extending LMMs to video understanding (Zhang et al., 2025a; Li et al., 2023b;c; Song et al., 2024; Fang et al., 2023a; Ataallah et al., 2024), including video captioning, question answering, and retrieval (Fang et al., 2023b). To support this development, a number of video benchmarks (Fu et al., 2024; Zhou et al., 2024; Li et al., 2024c; Wu et al., 2024; Chandrasegaran et al., 2024) have recently been proposed to enable more comprehensive evaluations of LMMs in dynamic visual environments.

However, existing video understanding evaluations primarily focus on major events, character actions, and interpersonal relationships, while largely overlooking video text. As a self-descriptive visual component, text in videos plays a crucial role in visual understanding (Zhang et al., 2025b; Zhao et al., 2022; Xu et al., 2021). On one hand, it provides **explicit perceptual cues**, such as street signs, storefronts, or subtitles, that help identify key elements and clarify the scene. On the other hand, text also enables **contextual reasoning**, revealing underlying motivations or causal relationships. For example, a "SALE" sign in a shop may explain why people are gathering, which is not readily apparent from visual cues alone.

Compared to images, perceiving dynamic video text and modeling its interaction with evolving visual contexts in videos is significantly more challenging. It requires not only fine-grained local-

Table 1: Comparison of video understanding benchmarks. "Vid", "Cli" and "Ins" denote video-level, clip-level and instance-level tasks. "T", "S" and "C" mean temporal, spatial and causal dimensions. "MC" and "OE" represent multiple-choice and open-ended questions. "TQ": the percentage of questions containing visual text instances.

| Benchmark | Video | QA | TQ % | Multi Lingual | Multi Source | Multi Granularity | Perception | | Reasoning | | | Paired Tasks | TaskType |
|---|---|---|---|---|---|---|---|---|---|---|---|---|---|
| | | | | | | | T | S | T | S | C | | |
| *General Video Understanding Datasets* | | | | | | | | | | | | | |
| NExT-QA (Xiao et al., 2021) | 5,440 | 52,044 | – | ✗ | ✓ | Vid+Cli | ✗ | ✗ | ✓ | ✗ | ✓ | ✗ | MC+OE |
| MVBench (Li et al., 2024c) | 4,000 | 4,000 | – | ✗ | ✓ | Vid+Cli | ✓ | ✓ | ✓ | ✗ | ✓ | ✗ | MC |
| MovieChat-1K (Song et al., 2024) | 1,000 | 13,000 | – | ✗ | ✓ | Vid+Cli | ✗ | ✗ | ✓ | ✗ | ✗ | ✗ | MC+OE |
| Video-MME (Fu et al., 2024) | 900 | 2,700 | – | ✓ | ✓ | Vid+Cli+Ins | ✓ | ✗ | ✓ | ✗ | ✗ | ✗ | MC |
| MLVU (Zhou et al., 2024) | 1,730 | 3,102 | – | ✓ | ✓ | Vid+Cli+Ins | ✗ | ✗ | ✗ | ✗ | ✗ | ✗ | MC+OE |
| *Video Text Datasets* | | | | | | | | | | | | | |
| BovText (Wu et al., 2021) | 2,000 | – | – | ✓ | ✓ | Ins | ✓ | ✓ | ✗ | ✗ | ✗ | ✗ | OE |
| RoadText1k (Reddy et al., 2020) | 1000 | – | – | ✗ | ✗ | Ins | ✓ | ✓ | ✗ | ✗ | ✗ | ✗ | OE |
| M4ViteVQA (Zhao et al., 2022) | 680 | 2,103 | 40 | ✗ | ✓ | Cli+Ins | ✗ | ✗ | ✓ | ✓ | ✗ | ✗ | MC+OE |
| RoadTextVQA (Tom et al., 2023) | 329 | 1,052 | 60 | ✗ | ✗ | Cli+Ins | ✗ | ✗ | ✓ | ✗ | ✗ | ✗ | MC |
| EgoTextVQA (Zhou et al., 2025) | 1,507 | 7,064 | 52 | ✓ | ✗ | Cli+Ins | ✗ | ✗ | ✓ | ✓ | ✗ | ✗ | OE |
| **Ours** | 939 | 2,857 | 65 | ✓ | ✓ | Vid+Cli+Ins | ✓ | ✓ | ✓ | ✓ | ✓ | ✓ | MC+OE |

ization at the instance level, but also temporal tracking and spotting at the clip level, as well as holistic understanding at the video level. Furthermore, video text appears in a wide range of scenarios and across multiple languages, which further increases the complexity of recognition and reasoning. Based on these insights, we propose **VidText**, a comprehensive benchmark for video text understanding, which introduces the following key advantages:

- **It encompasses a wide variety of video genres**, including media, entertainment, ego-centric, sports, life record, and knowledge, with 27 fine-grained categories covering diverse scenarios rich in visual text, such as scene text and subtitles. Moreover, it includes multilingual content, covering English, Chinese, Korean, Japanese, and German.
- **It supports multi-granularity evaluation**, including video-level, clip-level, and instance-level tasks. Video-level tasks involve holistic OCR understanding and reasoning over global video content. Clip-level tasks are designed to require localized comprehension based on specific temporal segments. Instance-level tasks demand fine-grained temporal and spatial grounding of individual text instances to support precise question answering.
- **It spans from visual text perception to cross-modal reasoning with visual context**. Building upon the meticulously annotated video text data, we produce video text-centric Chain-of-Thought (CoT) annotations, explicitly capturing the reasoning process between video descriptions and embedded texts, including spatial relationships with surrounding objects and temporal dependencies related to actions or events. In this way, we extend video text perception tasks into their corresponding reasoning counterparts, forming a comprehensive paired perception–reasoning framework that spans eight tasks covering multiple levels of understanding.

Tab. 1 shows that VidText enables a more comprehensive evaluation of video text understanding compared to both general video understanding benchmarks and video text-specific benchmarks. We conduct extensive evaluations on 18 popular LMMs, revealing several important insights. First, *video text understanding remains a technically challenging task for existing models*. Although Gemini 1.5 Pro (Team et al., 2023) achieves the highest performance, it only reaches an average score of 46.8%, and all models perform poorly on multi-granularity tasks, which is far below estimated human-level performance. Second, several concurrent open-source models (Zhang et al., 2025a; Chen et al., 2024b) demonstrate competitive performance, narrowing the gap with proprietary systems. Third, our empirical findings prove the design principles of the benchmark, as well as exploring the crucial factors on video text understanding, including OCR capability, input resolution, auxiliary information, and Chain-of-Thought strategies.

## 2 RELATED WORK

### 2.1 VIDEO LARGE LANGUAGE MODELS

With the rapid advancement of large language models (LLMs), a series of video large language models (Video LLMs) have emerged (Liu et al., 2023b; Zhu et al., 2023; Liu et al., 2023a; Chen et al., 2024c), leveraging LLMs as backbones to enhance video reasoning capabilities. Early Video

LLMs primarily relied on sparsely sampled frames and temporal modeling mechanisms (Li et al., 2023a; Liu et al., 2024d), such as Q-Former and temporal pooling, to facilitate video captioning and question answering. Building upon these designs, subsequent models (Bai et al., 2025; Chen et al., 2024d; Zhang et al., 2025a; Li et al., 2024d; Liu et al., 2024a; Zhang et al., 2024; Shu et al., 2024; Liu et al., 2025; Yuan et al., 2025) have focused on addressing key challenges in video understanding, including fine-grained semantic alignment, temporal representation, and long-duration video comprehension. For instance, Qwen-VL 2.5 (Bai et al., 2025) introduces dynamic resolution processing and absolute temporal encoding to handle variable-resolution videos. Video-LLaMA3 (Zhang et al., 2025a) applies a frame compression strategy based on frame similarity to reduce the number of visual tokens, resulting in more compact and precise video representations. To handle extremely long videos, LongVA (Zhang et al., 2024) extends the context length of the LLM backbone and transfers its long-context capability to the video domain. Video-XL (Shu et al., 2024) leverages the inherent key-value sparsification mechanism of LLMs to efficiently condense visual inputs. VideoChatFlash (Li et al., 2024d) proposes a hierarchical compression strategy, reducing token redundancy in both the video and language modules.

## 2.2 Video Understanding Benchmarks

With the growing interest in video LLMs, the development of dedicated benchmarks has become increasingly emphasized. Existing benchmarks have been designed for a variety of video understanding tasks, including action reasoning, spatio-temporal inference, video captioning, and long-video comprehension (Fu et al., 2024; Zhou et al., 2024; Li et al., 2024c; Wu et al., 2024; Xiao et al., 2021; Patraucean et al., 2023; Liu et al., 2024b; Maaz et al., 2023). For example, NeXT-QA (Xiao et al., 2021) evaluates temporal reasoning abilities by testing models on the relationships between human actions. VideoChatGPT-Bench (Maaz et al., 2023) focuses on open-ended video conversation, constructing captioning and dialogue tasks to assess generative and interactive capabilities. TempCompass (Liu et al., 2024b) introduces fine-grained temporal perturbations to assess whether models can answer questions based on temporal changes within the video. To support comprehensive video question answering, MVBench (Li et al., 2024c) proposes a large-scale benchmark covering 20 distinct subtasks, spanning multiple perception and reasoning dimensions. For long-video understanding, VideoMME (Fu et al., 2024), MLVU (Zhou et al., 2024), LVBench (Wang et al., 2024b) and LongVideoBench (Wu et al., 2024) curate diverse and extended-duration videos to evaluate multi-level abilities across extended temporal contexts.

As text carries rich and structured information in videos, several benchmarks have been proposed to evaluate video text understanding (Zhao et al., 2022; Wu et al., 2021; Reddy et al., 2020; Zhou et al., 2025; Wu et al., 2023), including tasks such as text tracking, spotting, and reasoning. Specifically, RoadTextVQA (Reddy et al., 2020) focuses on autonomous driving scenarios, while EgoTextVQA (Zhou et al., 2025) targets egocentric perspectives in daily life settings. In addition, M4-ViteVQA (Zhao et al., 2022) collects videos from nine diverse real-world scenarios, such as shopping, traveling, and movies, to evaluate the generalization capabilities of video-language models. However, these benchmarks exhibit two notable limitations. First, their task types are relatively simple, and therefore insufficient for comprehensively evaluating the diverse capabilities of modern video LLMs. Second, their video categories and language coverage remain limited, often constrained to specific application domains.

## 3 Dataset Construction

In this section, we describe the dataset construction process for VidText. We begin by illustrating how the source videos are collected (Sec. 3.1), followed by a detailed explanation of the annotation pipeline (Sec. 3.2). Finally, we describe the task taxonomy of our benchmark (Sec. 3.3).

### 3.1 Video Collection

In VidText, we aim to evaluate video text understanding across diverse scenarios, including both video category variety and language diversity. While several existing datasets (Zhao et al., 2022; Wu et al., 2021; Reddy et al., 2020; Wu et al., 2023) provide detailed text annotations, they all suffer from several key limitations: (1) Limited scenario diversity: Most datasets focus on specific

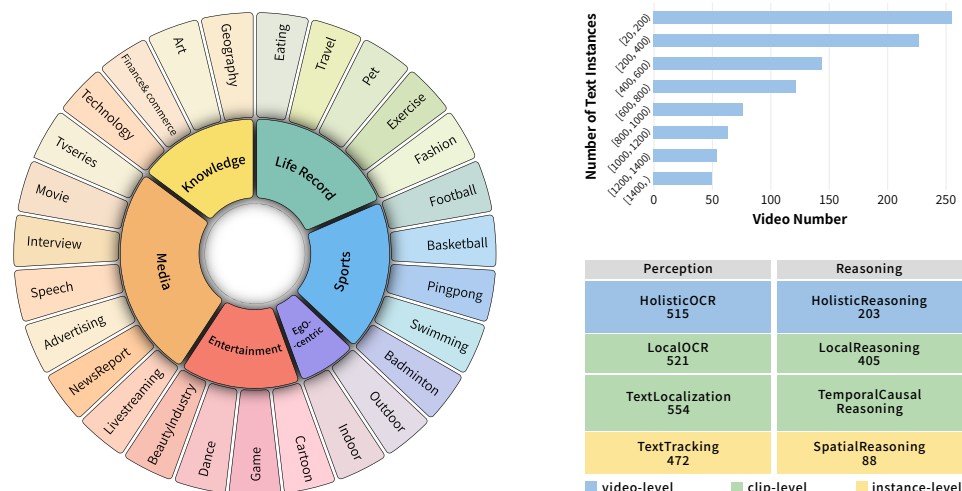

Figure 1: Statistical overview of our VidText. **(Left)** Video genres included in VidText. **(Top Right)** Visual Text Instance Distribution. **(Bottom Right)** Hierarchical Task type settings.

domains such as indoor scenes or egocentric videos, lacking coverage of richer, more interactive contexts such as sports events, livestreaming games, or daily vlogs. (2) Lack of language diversity: Nearly all existing datasets contain only English, failing to reflect the multilingual nature of real-world video text. (3) Short video duration: Many videos are only 10–15 seconds long, which limits their suitability for tasks involving cross-temporal reasoning or holistic understanding.

Therefore, in addition to incorporating existing datasets, we further collect video data from comprehensive long-form video benchmarks (Fu et al., 2024; Zhou et al., 2024) and public platforms such as YouTube, in order to enhance the scenario diversity, temporal richness, and linguistic coverage.

For the manually collected videos, we leverage expert models to construct an effective selection pipeline. First, we ensure the presence of visual text in each video by using Gomatching (He et al., 2024), a video text detection tool, to assess text density. Second, we filter out low-quality videos containing blur, watermarks, or low resolution, using existing video quality assessment models (Wen et al., 2024; Mi et al., 2024). Third, we enforce a minimum duration threshold of 3 minutes to guarantee sufficient temporal content. As a result, we collect a total of 939 high-quality videos, each annotated with one of 27 predefined scene categories. Additionally, we record metadata for each video, including language type, resolution, frame rate, and text density. Fig. 1 presents basic statistics of VidText. More detailed statistics across multiple dimensions are provided in the Supplementary Materials.

## 3.2 ANNOTATIONS GENERATION

To support evaluation at both the perception and reasoning levels, VidText provides meticulously constructed annotations tailored to the requirements of each task.

**Perception.** For each qualified video, we adopt a bottom-up strategy to construct multi-granularity annotations, including instance-level, clip-level, and video-level information. First, annotators are instructed to track at least three clear visual text instances throughout the video. For each instance, we conduct frame-by-frame fine-grained annotation until it disappears, generating a sequence of annotations that include bounding boxes, transcriptions, and unique track IDs. Second, the video is segmented into multiple intervals based on its duration (i.e., longer videos are divided into more segments). For each segment, we check the presence of visual text using instance-level annotations and record clip-level labels, including the temporal span (start and end timestamps) and associated transcriptions. Third, a separate group of annotators performs video-level annotations, which involve recording all distinct transcriptions that appear across the entire video. Specifically, for Chinese, we use text lines as the basic annotation unit, while for other languages, annotations are performed at the word level.

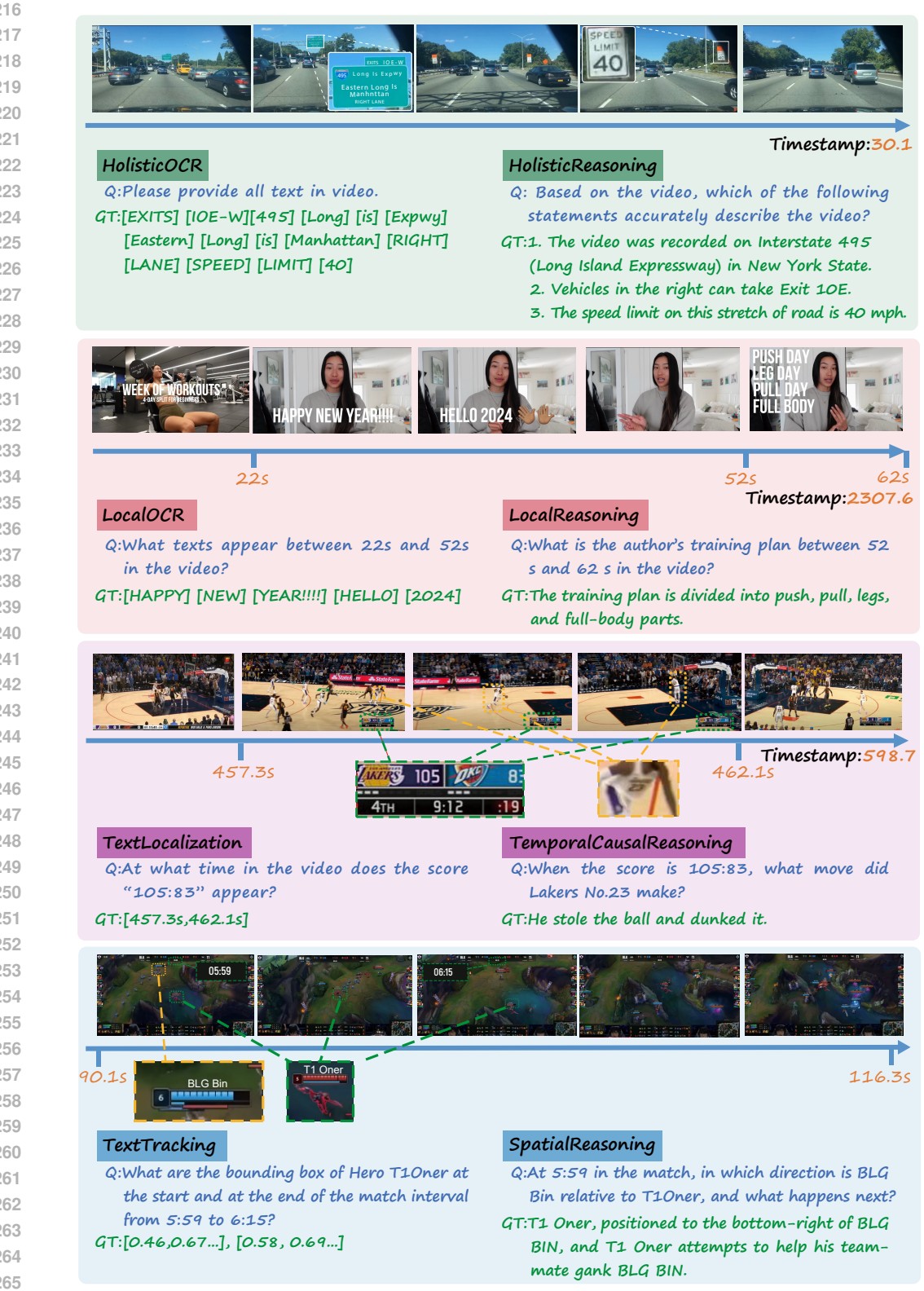

Figure 2: Examples from VidText. The benchmark includes eight tasks, featuring paired perception and reasoning components designed to evaluate the *video-level*, *clip-level*, and *instance-level* capabilities of LMMs. Given the video input and textual prompt, models are required to solve the tasks, with ground-truth answers highlighted in green.

**Reasoning.** Since the multi-granularity annotations constructed for perception address the question of "what texts appear in the video or clip", we further investigate "what actions or events are linked to these texts". To this end, we design a video text-centric Chain-of-Thought (CoT) annotation pipeline. First, for each video or clip (as defined by the time span annotations), we apply an adaptive sampling strategy to extract key frames. Then, we utilize the powerful vision-language model Aria (Li et al., 2024b) to generate high-quality frame-level captions, capturing both intra-frame and inter-frame contextual information. Based on the paired OCR transcripts and the multimodal descriptions, human annotators are instructed to design QA pairs that focus on the semantic or causal relationships between visual text and surrounding visual content. To ensure the quality of reasoning QA pairs, we enforce two post-validation principles: (1) Mask the visual texts and verify whether the question can be answered using only the visual content; (2) Mask the visual frames and check whether the question can be answered using only the textual information.

### 3.3 TASK TAXONOMY

Based on the detailed annotations encompassing perception and reasoning, we further define 8 hierarchical tasks which are demonstrated in Fig. 2.

**Holistic OCR & Holistic Reasoning.** Holistic OCR requires the model to recognize all visual texts appearing throughout the entire video. Redundant entries are removed, and the remaining text instances are sorted in chronological order. We evaluate this task using the F1-score, which is calculated based on instance-level precision and recall. Holistic Reasoning assesses the model's ability to understand the overall topic of the video by integrating recognized textual information with global semantic context. The task is formulated as a multi-label selection problem, where the model is asked to choose three correct answers from seven candidate options. Performance is measured by top-3 accuracy.

**Local OCR & Local Reasoning.** In contrast to holistic tasks, Local OCR and Local Reasoning focus on the model's ability to spot and interpret visual text within user-specified video segments. Local OCR requires recognizing all visual texts that appear within a given segment and is evaluated using the F1-score based on instance-level matching. Local Reasoning assesses the model's ability to infer local semantic meaning or intent from the text. It is formulated as a multiple-choice question, and performance is measured by answer accuracy.

**Text Localization & Temporal Causal Reasoning.** Similar to temporal grounding tasks, Text Localization requires the model to accurately predict the temporal interval during which a specific text appears in the video. The task is evaluated using Mean Intersection-over-Union (mIoU) based on ground-truth temporal spans. The corresponding reasoning task, Temporal Causal Reasoning, extends beyond localization to assess whether the model can infer causal relationships between identified texts and subsequent multimodal events or actions. Standard evaluation is conducted in a multiple-choice format, with accuracy as the performance metric.

**Text Tracking & Spatial Reasoning.** Given a target text instance, Text Tracking requires the model to predict its spatial bounding box locations at its first and last appearance within the video. Spatial Reasoning extends this task by asking the model to infer spatial relationships between the textual instance and surrounding visual elements at a specified timestamp. To enable standardized evaluation with LMMs, both tasks are formatted as multiple-choice questions.

## 4 EXPERIMENTS

### 4.1 SETTINGS

We conduct a comprehensive evaluation of 18 large multimodal models (LMMs) using our VidText benchmark, encompassing both open-source and proprietary models. For proprietary models, we evaluate the Gemini series (Team et al., 2023) and GPT series (Achiam et al., 2023; Hurst et al., 2024), using their official multi-image evaluation APIs. For open-source models, we select current state-of-the-art video LMMs with diverse architectures and LLM sizes, enabling a broad assessment of video text understanding capabilities. All evaluations are conducted in a zero-shot manner. More details about the evaluation settings are provided in the Supplementary Materials.

Table 2: The overall performance on VidText. **HoliOCR**: Holistic OCR; **HoliRea.**: Holistic Reasoning; **LocalOCR**: Local OCR; **LocalRea.**: Local Reasoning; **TextLocal.**: Text Localization; **TempCauRea.**: Temporal Causal Reasoning; **TextTrac.**: Text Tracking; **SpaRea.**: Spatial Reasoning; **Avg.**: the average performance of the eight tasks. The best Accuracy / Score results are highlighted.

| Method | Size | Avg. | HoliOCR | HoliRea. | LocalOCR | LocalRea. | TextLocal. | TempCauRea. | TextTrac. | SpaRea. |
|---|---|---|---|---|---|---|---|---|---|---|
| *Human* | – | 89.5 | 92.8 | 96.0 | 94.3 | 95.7 | 81.3 | 88.6 | 80.3 | 87.3 |
| *proprietary LMMs* | | | | | | | | | | |
| GPT-4-Turbo (Achiam et al., 2023) | – | 29.7 | 22.9 | 28.7 | 36.7 | 36.5 | 15.8 | 39.4 | 24.3 | 33.6 |
| Gemini 1.5 Flash (Team et al., 2023) | – | 34.7 | 26.3 | 34.0 | 40.2 | 42.4 | 28.9 | 40.0 | 30.7 | 35.4 |
| GPT-4o (Hurst et al., 2024) | – | 40.2 | 29.5 | 38.9 | 46.0 | 43.3 | 45.5 | 42.5 | 36.2 | 39.8 |
| Gemini 1.5 Pro (Team et al., 2023) | – | **45.3** | **34.8** | **43.6** | **50.2** | **50.1** | **48.7** | **47.0** | **40.3** | **47.9** |
| *Open-source LMMs* | | | | | | | | | | |
| LongVU (Shen et al., 2024) | 3B | 17.0 | 5.8 | 20.4 | 15.4 | 17.0 | 15.6 | 15.9 | 15.4 | 30.5 |
| Qwen2.5-VL (Bai et al., 2025) | 3B | 21.1 | 11.4 | 23.2 | 28.5 | 17.8 | 18.7 | 15.4 | 18.3 | 35.3 |
| Video-XL-Pro (Liu et al., 2025) | 3B | 22.5 | 10.9 | 22.9 | 30.4 | 15.6 | 18.7 | 27.9 | 20.9 | 32.9 |
| LongVA (Zhang et al., 2024) | 7B | 19.2 | 4.8 | 5.6 | 3.2 | 46.9 | 4.5 | 28.3 | 29.6 | 30.5 |
| MiniCPM-V2.6 (Yao et al., 2024) | 7B | 26.5 | 29.2 | 21.2 | 11.4 | 42.9 | 13.3 | 30.3 | 20.5 | 43.2 |
| VideoChatFlash (Li et al., 2024d) | 7B | 29.2 | 13.6 | 13.3 | 1.0 | 50.1 | 45.1 | 42.4 | 23.3 | 44.3 |
| Qwen2-VL(Wang et al., 2024a) | 7B | 30.3 | 27.0 | 34.0 | 37.5 | 23.7 | 11.2 | 42.4 | 24.6 | 42.1 |
| Qwen2.5-VL (Bai et al., 2025) | 7B | 31.9 | 35.9 | 36.0 | 37.0 | 26.5 | 26.5 | 35.4 | 22.4 | 35.2 |
| VideoLLaMA3 (Zhang et al., 2025a) | 7B | **39.9** | 23.5 | 31.5 | 39.2 | 41.2 | **47.3** | **55.6** | 31.1 | 50.0 |
| ShareGPT4Video (Chen et al., 2024a) | 8B | 16.4 | 2.5 | 2.6 | 0.8 | 43.5 | 0.0 | 27.3 | 28.0 | 26.1 |
| Oryx-1.5 (Liu et al., 2024d) | 32B | 35.4 | 35.3 | 33.9 | 30.8 | 48.5 | 26.7 | 45.2 | 26.0 | 36.4 |
| LLava-OV(Li et al., 2024a) | 72B | 36.1 | 20.1 | 28.1 | **41.3** | 49.4 | 9.9 | 54.6 | **31.8** | **53.4** |
| Qwen2.5-VL (Bai et al., 2025) | 72B | 38.5 | 40.1 | **49.3** | 35.9 | 28.2 | 28.7 | 52.5 | 31.1 | 42.1 |
| InternVL2.5 (Chen et al., 2024b) | 78B | 39.8 | **40.2** | 37.4 | 29.0 | **50.4** | 30.5 | 48.5 | 29.9 | 52.3 |

## 4.2 MAIN RESULTS

The overall evaluation results for all investigated LMMs in the VidText are shown in Tab. 2. Individual performances are reported for each task, while average performances are provided. From the results, we derive three primary conclusions:

1) **Gemini 1.5 Pro (Team et al., 2023) achieves the best performance on our benchmark**. It significantly outperforms other models on video-text-based perception and reasoning tasks.

2) **Proprietary models typically perform better than open-source models**. However, some open-source models deliver surprisingly strong results on specific tasks. For example, VideoLLaMA3 (Zhang et al., 2025a) achieves the highest performance on both Temporal Causal Reasoning and Spatial Reasoning.

3) **Video text understanding remains challenging for current video LMMs**. Current models fall far short of human-level performance, show limited ability in fundamental video OCR tasks (where specialized OCR models often outperform), and struggle with multimodal reasoning based on visual text cues—with all video multiple-choice reasoning tasks achieving below 60% accuracy, significantly lagging behind similar image-based tasks (Biten et al., 2019; Singh et al., 2019).

Beyond the primary conclusions on overall performance, we further analyze model behaviors across individual tasks.

4) **Among multi-granular tasks, video-level and instance-level tasks are more challenging than clip-level tasks, across both perception and reasoning settings.** We hypothesize that this is due to the limited capabilities of current LMMs in two aspects: video-level tasks require global information aggregation, while instance-level tasks demand fine-grained retrieval and grounding, both of which remain weak points for existing models.

5) **For video-level and instance-level tasks, the performance of perception and reasoning shows a strong correlation, while the two appear relatively independent in clip-level tasks.** This may be because certain clip-level perception tasks, such as text localization, require accurate temporal grounding based on fine-grained visual cues. However, the corresponding reasoning tasks, such as temporal reasoning, can often be solved using local visual clues from sparsely sampled frames, allowing models to bypass the need for precise perception outputs.

6) **Scaling up the size of LLMs leads to more significant performance gains on reasoning tasks compared to perception tasks.** This suggests that video text perception cannot be effectively improved by model scale alone, and instead requires careful architectural design, specialized training data, and other task-specific considerations.

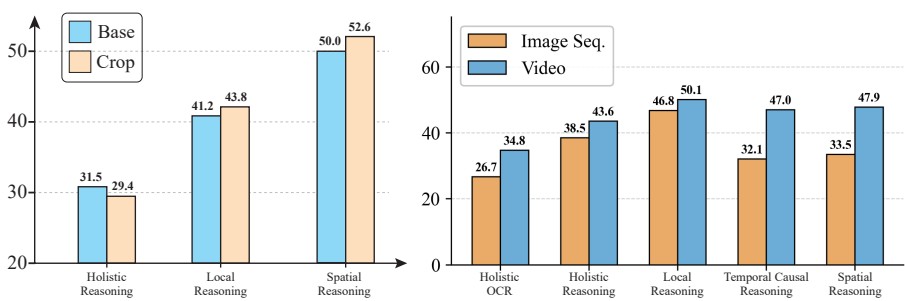

Figure 3: Ablations on the multi-granularity design (**left**) and temporal modeling (**right**) of VidText.

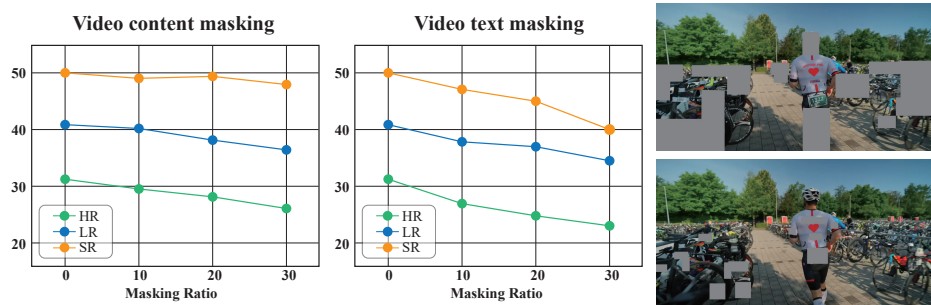

Figure 4: Ablation studies on the joint reasoning of video texts and video contents. "HR", "LR" and "SR" denote Holistic Reasoning, Local Reasoning and Spatial Reasoning, respectively. We visualize "Video content masking" and "Video Text masking" in the right part.

# 5 ABLATION STUDIES

## 5.1 INVESTIGATING THE EFFECTIVENESS OF VIDTEXT DESIGN

**Joint video text and multimodal contexts reasoning.** VidText successfully extends perception-level tasks into reasoning tasks, which require the joint modeling of video texts and their multimodal contextual information. To validate this, we perform an ablation study by selectively masking either the visual text regions or the surrounding video content at varying random ratios. As shown in Fig. 4, the performance on all reasoning tasks consistently drops as the masking ratio increases, confirming that both textual and visual cues are essential for reasoning under our task design.

**Multi-granularity design.** VidText includes multi-granular tasks spanning video-level, clip-level, and instance-level. To verify that tasks at different levels require correspondingly different levels of contextual information, we conduct ablation studies using VideoLLaMA3 (Zhang et al., 2025a). Specifically, for holistic tasks, we randomly extract 50% of the video duration as a segment and evaluate performance on Holistic Reasoning. For clip-level and instance-level tasks, we select key clips based on their original task annotations. As shown in Fig. 3, clip-level and instance-level tasks benefit significantly from segment-based evaluation, as key frames provide concentrated visual text information. In contrast, Holistic Reasoning performance declines, as the task requires global information aggregation, which is lost when only partial segments are used.

**Temporal Modeling.** A fundamental question in video-text understanding is whether tasks genuinely require video-level temporal modeling or whether processing sequential image inputs is sufficient. To address this question, we conduct a controlled comparison using Gemini 1.5 Pro under two experimental conditions: (1) *Video mode*, where the complete video is provided as a unified input; and (2) *Image-sequence mode*, where uniformly sampled frames are fed as a sequence of discrete images. As shown in Fig. 3, video input consistently outperforms sequential image across nearly all evaluated tasks. This performance gap indicates that integrated temporal modeling and redundancy suppression are essential components for effective video-text understanding. Notably, tasks requiring temporal reasoning exhibit the most substantial improvements, with Temporal Causal Reasoning and Spatial Reasoning showing the largest performance gains.

Table 3: Detailed analysis about the impact of input resolution, image OCR ability, and LLM Backbone. LLaVA-Next: LLaVA-Next-Video (Liu et al., 2024a).

| Impact of input resolution | | | Impact of OCR ability | | | Impact of LLM | | |
|---|---|---|---|---|---|---|---|---|
| Model | resolution | Avg | Model | OCRBench | Avg | Model | LLM | Avg |
| Oryx-1.5 | $448^2$ | 35.4 | LLaVA-OV | 621 | 36.1 | LLaVA-OV | Qwen2-7B | 22.1 |
| | $896^2$ | 38.6↑3.2 | VideoLLaMA3 | 828 | 39.9↑3.8 | | Qwen2-72B | 36.1↑14.0 |
| InternVL | $448^2$ | 39.8 | GPT4V | 645 | 29.7 | LLaVA-Next | LLaMA3-8B | 15.3 |
| | $896^2$ | 44.8↑5.0 | GPT-4o | 822 | 40.2↑10.5 | | Qwen2-7B | 20.8 ↑5.2 |

Table 4: Ablations about auxiliary information (**Left**) and CoT strategy (**Right**) for video text understanding. "HR", "LR", "TR" and "SR" denote Holistic Reasoning, Local Reasoning, Temporal Causal Reasoning and Spatial Reasoning.

| Method | HR | LR | TR | SR |
|---|---|---|---|---|
| Qwen2.5-VL | 36.0 | 26.5 | 35.4 | 35.2 |
| Qwen2.5-VL + Audio | 36.3 | 26.6 | 35.2 | 35.4 |
| Qwen2.5-VL + Text | 37.2 | 28.3 | 37.9 | 38.1 |
| Qwen2.5-VL + Audio + Text | **37.6** | **29.5** | **38.0** | **39.5** |

| Method | HR | LR | TR | SR |
|---|---|---|---|---|
| Qwen2.5-VL | 36.0 | 26.5 | 35.4 | 35.2 |
| Qwen2.5-VL + CoT | **40.5** | **28.7** | **37.2** | **40.9** |
| VideoLLaMA3 | 31.5 | 41.2 | 55.6 | 50.0 |
| VideoLLaMA3 + CoT | **33.8** | **44.6** | **56.2** | **53.8** |

## 5.2 EXPLORING CRUCIAL FACTORS OF VIDEO TEXT UNDERSTANDING

**Model-intrinsic Factors.** As shown in Tab. 3, we conduct ablation studies on several factors. First, we examine the *impact of input resolution* using two representative models, Oryx-1.5 (Liu et al., 2024d) and InternVL2.5 (Chen et al., 2024d), both of which support adjustable input sizes. Increasing the resolution significantly improves video text understanding performance, especially in InternVL2.5 (Chen et al., 2024d), where the input images are divided into sub-patches to allow better preservation of text details. Second, to assess the *role of OCR capability*, we refer to each model's performance on standard OCR benchmarks such as OCRBench (Liu et al., 2024c). The results show that a model's video text understanding performance generally aligns with its fundamental OCR accuracy. Finally, we compare *different LLM backbones* and find that certain architectures (e.g., Qwen2.5) exhibit stronger performance in multilingual scenarios, often outperforming LLaMA-based variants. These observations collectively indicate that video text understanding is influenced by a combination of input fidelity, OCR capacity, and language modeling strength.

**External Factors.** As shown in Tab. 4, we first investigate whether external auxiliary information can enhance video text understanding, particularly for reasoning tasks. In this study, we consider audio transcripts and video text (e.g., subtitles or OCR outputs), both of which can be extracted using specialized tools. We convert these modalities into textual sequences and append them to the original query as contextual subtitles. As shown in our experiments, both sources contribute positively to performance. Video text provides stronger gains in global tasks that require long-range context, while audio transcripts are more beneficial for local tasks, possibly due to their alignment with short-term actions or events. Second, we propose a video text-centric Chain-of-Thought (CoT) reasoning strategy, which decomposes complex reasoning processes into structured sub-steps. Specifically, the video is uniformly segmented into multiple clips. For each clip, the model is prompted to: (1) spot all visible texts, (2) generate a detailed description of the clip, and (3) infer whether any visual texts are semantically related to the description and answer the reasoning question accordingly. This CoT-based prompting strategy yields consistent improvements across all reasoning tasks, highlighting the potential of test-time reasoning augmentation for video-language models.

## 6 CONCLUSION

This paper presents VidText, a comprehensive benchmark for evaluating video text understanding in large multimodal models. Through broad scenario coverage, multi-granular evaluation, and paired perception-reasoning tasks, VidText enables systematic analysis of LMM capabilities. Our empirical studies reveal that current LMMs face significant challenges in both perceiving and reasoning over video texts, requiring joint optimization of model-intrinsic factors (input resolution, OCR capability, LLM backbone) and external strategies (auxiliary modality integration, Chain-of-Thought prompting). We expect VidText to advance research in OCR and video understanding communities.

ETHICS STATEMENT

**Scope and Intended Use.** *VidText* is a research-only benchmark for **video text perception and reasoning** (e.g., scene texts, subtitles, signage). It is *not* designed for identity recognition, person tracking, surveillance, or any privacy-invasive applications. The dataset, code, and evaluation scripts are released to advance transparent, verifiable research on video–language understanding.

**Data Sources and Licensing.** VidText aggregates publicly-available video–text resources and established benchmarks under their original research licenses. We additionally include a curated set of public YouTube broadcasts (sports/esports). Redistribution is limited to **non-commercial academic research**. Items with ambiguous or restrictive licensing were excluded. For platform-sourced videos, we primarily release **indices and retrieval scripts** rather than raw videos; if research exemplars are strictly necessary, they are provided in **de-identified, downsampled, and sparsely sampled** form to reduce re-use risks.

**Consent and Human Involvement.** A small-scale human evaluation involved **three trained graduate RAs** who received an Information Sheet, provided **informed consent**, and were compensated at the institution's standard rate ($25/hour). No demographic attributes were collected, and no audio/video recordings of annotators were made. All annotation artifacts are stored on **encrypted drives** with access restricted to the author team. Consistent with US 45 CFR 46, EU GDPR, and our institution's policy, this activity qualifies as **Exempt / Not Human-Subjects Research (Category 4: publicly available / anonymized data)**.

**Privacy and De-identification.** We do not actively record new data. For public videos, potentially identifiable faces are **blurred**; channel identifiers/watermarks are **masked or cropped**; OCR/ASR textual outputs that may contain **personally identifiable information (PII)** are removed or replaced with placeholders. We perform a combined rules-based and model-aided check before release, keep spot-check logs, and update the release when residual issues are reported.

**Copyright and Takedown.** We will provide a dedicated takedown contact on the project page and repository. Upon verified request from content owners or data subjects, we will **remove the corresponding content within 48 hours**. When removal is necessary, we follow established precedents by retaining only **sparse frames** or **annotation/metadata** indispensable for scientific reproducibility. We keep audit logs of takedown requests and actions.

**Bias, Fairness, and Transparency.** The Dataset Card reports **language/region distributions**, task coverage, and per-subset performance to surface disparities. We encourage reporting per-subset metrics and conducting follow-up robustness/fairness analyses.

**Misuse Prevention.** The license and documentation **explicitly prohibit** surveillance, biometric identification, targeted profiling, and other privacy-invasive or unlawful uses. Redistribution of original platform videos is discouraged; instead, we provide indices and scripts that respect platform Terms of Service (ToS).

**Environmental Considerations.** We report hardware profiles and approximate GPU-hours for key experiments. To reduce energy cost, we offer evaluation pipelines with **frame sampling**, **resolution caps**, and **cache reuse**, and we avoid redundant inference in ablations.

**Disclaimer.** VidText is released solely for non-commercial academic research. Users must ensure compliance with applicable laws and data-source ToS; the authors do not grant rights beyond the stated license. Verified complaints will be processed within 48 hours according to our takedown policy.

REPRODUCIBILITY STATEMENT

**Release Artifacts.** We release: (i) JSON schemas and examples for all VidText tasks (video-/clip-/instance-level); (ii) data indices and retrieval scripts (plus SHA256 checksums and directory

layout); (iii) unified *scoring scripts* for every task; (iv) one-command evaluation/ablation scripts; (v) a Dockerfile and `conda` environment specification; and (vi) logs for key experiments (hardware profile, GPU-hours). Public entry points and documentation are provided at `<repo_url>`.

**Evaluation Protocol.**  Each task specifies metric definitions and a single entry script. We **fix** the following to avoid hidden degrees of freedom: sampling rate (FPS), temporal window and stride, maximum frames per sample, input-resolution caps, and the concatenation rules for auxiliary text (OCR/ASR). Default seeds and decoding parameters are provided; any deviations must be reported.

**Models and Checkpoints.**  We list external models and versions with exact checkpoint identifiers (e.g., `Video-LLaMA3-<ckpt>`, `InternVL2.5-<ckpt>`, `Qwen-VL-2.5-<ckpt>`, `Gemini-1.5-Pro-<date>`). Inference configs include: input resolution, max token/sequence length, sampling temperature/top-$p$, and *video-text-centric Chain-of-Thought (CoT)* prompt templates. All prompts are released verbatim.

**Auxiliary Pipelines (OCR/ASR).**  We provide reproducible OCR/ASR pipelines with engine versions and settings: OCR (engine name/version, language packs, page segmentation/reading order options) and ASR (engine name/version, sampling rate, VAD settings). A standardized *prompt-concatenation template* appends cleaned OCR/ASR text to user queries, preserving determinism.

**Ablation Protocols (One-Command).**  We supply toggles and single-entry scripts for the following, each with fixed hyperparameters:

- **Temporal modeling:** establishes the necessity of true video input.
- **Multi-Video Effectiveness:** cross-clip evidence aggregation (e.g., voting, retrieval-and-rerank).
- **Resolution Scans:** resolution sweep (e.g., {224, 336, 448, 560}) and visual partition settings.
- **OCR Capability Alignment:** alignment table mapping core OCR scores to downstream performance.
- **Aux/CoT  On–Off:**  `visual-only`, `visual+OCR`, `visual+ASR`, `visual+OCR+ASR`, `visual+CoT`, `visual+Aux+CoT`.

**Canonical Commands.**  We expose minimal, deterministic entry points; actual script names are provided in the repo README.

**Determinism and Environment.**  We fix random seeds across Python/NumPy/PyTorch and enable deterministic backends when available. We document unavoidable non-deterministic sources (e.g., vendor kernels, closed APIs). We provide a Docker image and a `conda env.yml` with exact package versions, CUDA/cuDNN/driver requirements, and OS details.

**Hardware Disclosure.**  We report GPU models/counts, driver/CUDA/cuDNN versions, average inference time, and peak memory for principal baselines and ablations. Where applicable, we report energy-related proxies (e.g., GPU-hours). Representative configs: `<GPU_type>`, `<CUDA_ver>`, `<driver_ver>`.

**Data Integrity and Versioning.**  All downloadable artifacts have SHA256 checksums. Releases are versioned; metric-affecting changes are accompanied by a revised reference evaluation. Takedown-driven updates (e.g., removal or de-identification improvements) are recorded in a public changelog.

**Support and Questions.**  Reproducibility questions and bug reports can be filed on the issue tracker or sent to our e-mail. We aim to respond within two business days.

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

# A  OVERVIEW OF APPENDIX

- **Limitations** (B)
- **Discuss** (C)
- **Use of LLMs** (D)
- **Broader impact** (E)
- **More Details of VidText** (F)
- **More ablation studies** (G)
- **Collecting details of VidText** (H)
- **Details of annotation** (I)
- **Detailed experimental results** (J)
- **Model prompts** (K)
- **More visualization results** (L)
- **Ethics and Responsible Dataset Use** (M)

# B  LIMITATIONS

We summarize the limitations of our work as follows:

- **Limited scenario coverage:** Although VidText includes 27 fine-grained video categories, it still lacks representation of long-tail or high-risk domains such as medical emergencies, industrial workflows, or disaster scenarios.
- **Imbalanced language distribution:** The majority of samples are in English and Chinese, with significantly fewer examples in other languages such as German, Korean, and Japanese. This imbalance prevents a thorough evaluation of multilingual OCR and reasoning capabilities.
- **Scarcity of challenging text instances:** VidText contains relatively few examples involving difficult text conditions such as severe occlusion, low resolution, motion blur, unusual fonts, or multi-line arrangements. This limits the benchmark's ability to fully assess model robustness under real-world noise and distortion.

# C  DISCUSSION

These dataset and model limitations are mutually reinforcing. Dataset gaps may conceal important weaknesses in current models, while existing models' deficiencies highlight the need for broader and more diverse benchmarks. Future efforts should focus on expanding long-tail scene and language coverage in VidText, while also improving LMM architectures with better multilingual OCR, noise robustness, and cross-modal reasoning abilities. Furthermore, we also summarize three insights as follows:

- **Weak cross-domain transfer:** Most LMMs are pretrained on image-based OCR tasks and struggle to generalize to unseen video scenes, such as sports broadcasts or livestream interfaces, where text appearance and context are highly dynamic.
- **Insufficient multilingual alignment:** Current models show limited ability in detecting, transcribing, and semantically linking non-English texts to the visual context, resulting in degraded performance on multilingual content.
- **Low robustness to visual noise:** Models often fail when confronted with noisy, blurry, or occluded text, particularly in tasks requiring instance-level grounding. This degrades downstream reasoning performance and reflects a need for stronger visual resilience.

# D  USE OF LARGE LANGUAGE MODELS (LLMS)

**Scope.**  We used LLMs *only for language polishing and light scripting assistance*. LLMs were not used to generate data, labels, model outputs, evaluation results, or figures.

**Writing assistance.**    LLMs were employed to improve grammar, wording, and clarity of prose and to reformat LaTeX. All technical content (methods, formulas, hyperparameters, tables, and numbers) was authored and verified by the authors.

**Scripting assistance.**    LLMs helped draft boilerplate code such as command-line wrappers, dataset loaders, or small utilities (e.g., argument parsing, logging). All scripts were *manually reviewed, edited, and tested* by the authors before inclusion. Final experimental pipelines are specified in our repository and Reproducibility Statement.

**No synthetic labels or data.**    No LLM-generated text was used as ground-truth labels, dataset entries, or to augment training/evaluation data. Test annotations, metrics, and reported results are human-authored or taken from public benchmarks per their licenses.

**Privacy and ToS.**    We did not upload private or license-restricted raw videos to third-party APIs. Any prompts contained no personally identifiable information. Our usage complies with data-source licenses and platform Terms of Service.

**Reproducibility.**    All scripts produced with LLM assistance are fully documented and version-controlled; seeds, hyperparameters, and command entry points are fixed. Therefore, the use of LLMs does not affect experimental validity or reproducibility.

## E    BROADER IMPACT

The VidText benchmark is poised to make a significant contribution to both the OCR and video understanding communities by bridging the gap between low-level text perception (Shu et al., 2025; Li et al., 2024e; Zeng et al., 2024) and high-level semantic reasoning (Long et al., 2021; Zhu et al., 2016) in video contexts.

For the OCR community, VidText offers a valuable opportunity to move beyond traditional image-based text detection and recognition (Huang et al., 2022; Zhou et al., 2017; Liao et al., 2020; Shu et al., 2023). By shifting the focus to temporal and contextual dynamics in videos, it promotes the development of algorithms that can track, ground, and interpret visual texts over time.

For the video understanding community, VidText introduces the underexplored yet semantically rich modality of scene text into the landscape of video-language research. By incorporating fine-grained text perception tasks and their paired reasoning counterparts, VidText pushes video-language models to integrate visual texts with multimodal contextual cues, fostering more explainable, interpretable, and grounded video understanding.

## F    MORE DETAILS OF VIDTEXT

**Scene and language distributions.**    Fig. 5 illustrates the distribution of visual text quantity across six video scene categories. The largest number of text instances appears in **Entertainment** and **Sports**-related content, while **Knowledge** and **Media** are less dense in text content.For completeness, Tab. 5 reports the proportional breakdown of languages. The two largest—**English** and **Chinese**—account for **46.1%** and **32.3%** of the corpus, respectively, while **Japanese (8.2%)**, **Korean (7.0%)**, and **German (6.4%)** together make up the remaining **21.6%**. This skew toward high-resource languages suggests that models may generalize better on English/Chinese content, whereas performance on lower-resource languages could be constrained by data scarcity.

**Video duration distribution.**    VIDTEXT exhibits a wide range of video durations, with an average length of 108.2 seconds. As shown in Fig. 6, this highlights the multi-duration characteristic of VIDTEXT, ensuring the temporal diversity needed to support both short-form and long-form video understanding tasks.

**Semantic content word cloud.**    To visualize the semantic richness and diversity of video–text interactions, we construct a word cloud using all questions and answers in VIDTEXT. As shown

Table 5: Distribution of scene super-categories and languages in VIDTEXT.

| Scene super-category | # Videos | Proportion |
|---|---|---|
| Media | 192 | 20.4% |
| Knowledge | 164 | 17.5% |
| Life-record | 180 | 19.2% |
| Entertainment | 123 | 13.1% |
| Ego-centric | 101 | 10.8% |
| Sport | 179 | 19.0% |
| **Languages** | | |
| English | 433 | 46.1% |
| Chinese | 303 | 32.3% |
| Japanese | 77 | 8.2% |
| Korean | 66 | 7.0% |
| German | 60 | 6.4% |

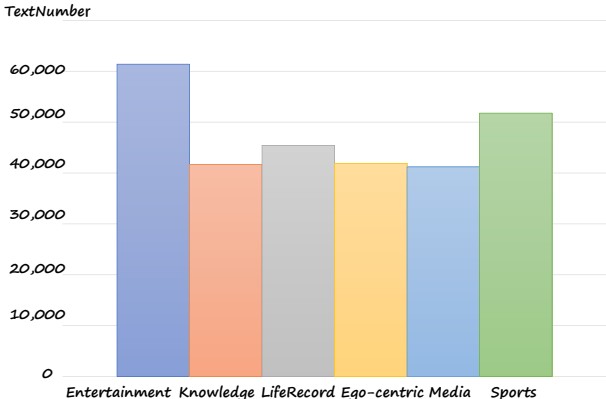

Figure 5: Text quantity distribution across six scene categories.

in Fig. 7, high-frequency words such as *text*, *video*, *content*, and *EXIT* reflect a strong alignment between text and semantic reasoning. The co-existence of spatial keywords (e.g., *LEFT*, *RIGHT*), functional terms (e.g., *score*, *speed*), and contextual references (e.g., *player*, *talent*) highlights the multi-granular reasoning needs of the dataset.

## G    MORE ABLATION STUDIES

### G.1    FURTHER IMPACT ANALYSIS OF KEY DESIGN CHOICES

**Impact of Video Duration**    To investigate the influence of video duration on various tasks, we grouped videos into five duration intervals and evaluated four representative tasks: Text Localization, Temporal Causal Reasoning, Local OCR, and Local Reasoning. The results are shown in Tab. 6.

**Observation.** Perception-heavy tasks (e.g., Text Localization and OCR) suffer a significant performance drop as video length increases. This indicates the challenge of long-range visual-text dependency modeling. Reasoning tasks also degrade but show more fluctuation. In particular, Temporal Causal Reasoning performs unexpectedly poorly on 30–60 s videos.

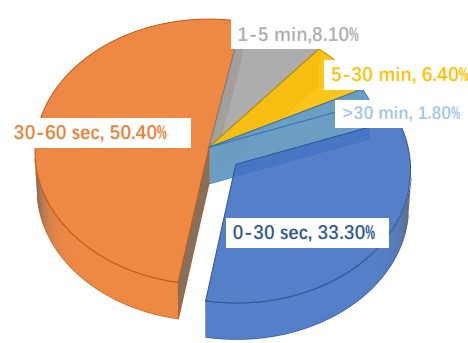

Figure 6: Video duration distribution in VIDTEXT.

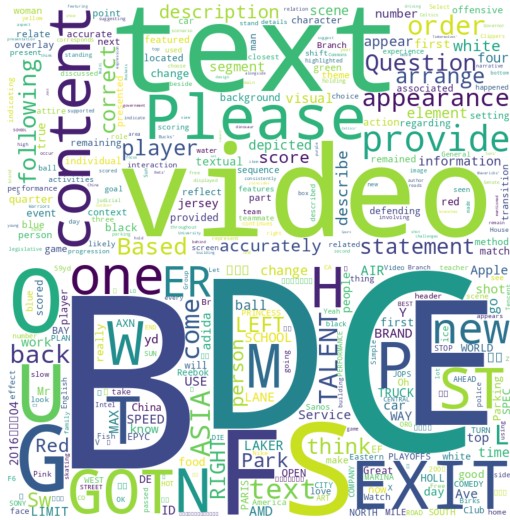

Figure 7: Word cloud of all questions and answers in VIDTEXT.

We hypothesize this may not only be due to duration, but also due to difficulty in locating the relevant temporal segment where the causal text appears. To verify this, we conduct an additional ablation study: for each sample, we crop the ground-truth segment and feed only that segment into the model. The results in Tab. 6 demonstrate that models perform significantly better when provided with only the relevant segment, confirming that **text localization** is a key bottleneck.

This ablation emphasizes that improving temporal grounding and retrieval capabilities is essential for long-form causal reasoning tasks.

**Language diversity ablation.** We analyze Holistic/Local OCR across languages (Tab. 7) for Qwen2.5-VL-7B and VideoLLaMA3-7B. We observe **notable performance drops** on lower-resource languages (Korean, Japanese, German) versus high-resource ones (English, Chinese), suggesting bias from training data distributions.

**Alignment with training distribution (controlled fine-tuning).** Since most model developers do not disclose language ratios, we conduct a controlled study on LLaVA-One-Vision by incrementally adding multilingual OCR data during instruction tuning: Phase 1 (IC13+IC15; English), Phase 2 (+ReCTS+TextOCR; add Chinese), Phase 3 (+MLT17+MLT19; add Korean/Japanese/German). We

Table 6: Impact of video duration and segment cropping.

| Impact of Video Duration | | | | Impact of Segment Cropping on TCR | | | |
|---|---|---|---|---|---|---|---|
| Duration Range | Text Loc. ↑ | OCR ↑ | Reasoning ↑ | Model | Full Video | Segment | Δ |
| 0–30 s | 29.81 | 27.07 | 45.45 | VideoLLaMA3 | 43.33 | 56.67 | (↑13.34) |
| 30–60 s | 29.01 | 24.25 | 54.55 | Qwen2.5-VL | 26.67 | 33.33 | (↑6.66) |
| 1–5 min | 14.45 | 30.21 | 42.86 | | | | |
| 5–30 min | 8.28 | 18.31 | 46.57 | | | | |
| ≥30 min | 0.00 | 0.00 | 33.33 | | | | |

Table 7: OCR performance (%) across languages. Comparison between Qwen2.5-VL and VideoL-LaMA3 on local and holistic OCR tasks.

| Language | Qwen2.5-VL↑ | | VideoLLaMA3↑ | |
|---|---|---|---|---|
| | Local OCR | Holistic OCR | Local OCR | Holistic OCR |
| English | 49.3 | 45.2 | 45.5 | 32.1 |
| Chinese | 26.3 | 25.4 | 24.5 | 22.6 |
| Korean | 17.6 | 16.4 | 18.9 | 13.5 |
| Japanese | 22.8 | 23.1 | 23.1 | 18.3 |
| German | 14.5 | 12.4 | 14.8 | 11.2 |

keep the original video instruction data to preserve video ability. Tab. 8 (Holistic OCR) shows consistent gains with multilingual supervision, especially on lower-resource languages.

## G.2 MODEL-INTRINSIC FACTORS

We ablate three influential factors (Tab. 3): **(i) Input resolution**: Increasing resolution improves video-text performance, especially for models that partition inputs into sub-patches (e.g., In-ternVL2.5), which better preserve text details at higher resolutions. **(ii) OCR ability**: Performance aligns with standard OCR benchmarks (e.g., OCRBench), indicating that fundamental OCR competence is predictive for video text understanding. **(iii) LLM backbone**: Some backbones (e.g., Qwen2.5) show stronger multilingual performance than LLaMA-based variants, suggesting language modeling strength matters alongside perception.

## G.3 CHAIN-OF-THOUGHT (COT) STRATEGY: EFFECTIVENESS AND SCOPE

**Why it works.** CoT boosts performance mainly due to two factors: (i) it decouples complex video–text reasoning into two sub-problems, *visual text spotting* and *video content understanding*, allowing the model to focus on one aspect at a time; (ii) it mitigates the temporal aggregation bottleneck in current Video LLMs by segmenting long videos and enabling localized reasoning within clips. For global tasks like Holistic Reasoning, CoT processes long videos sequentially, clip by clip; for local tasks (e.g., Local Reasoning, Temporal Causal Reasoning), it transforms the search space from vision to text, using a retrieval-like strategy to locate relevant cues.

**Ablations.** We design two ablated CoT variants to understand its impact(Tab. 9: (1) *Partial-Task CoT*, where prompts only include either video summarization (VS) or visual text spotting (VTS); and (2) *Partial-Time CoT*, where the model reasons over the full video instead of clip-wise segments. Results show that the full CoT strategy outperforms both ablations, demonstrating that joint modeling of segmented temporal reasoning and dual-modality decomposition is essential for complex video text understanding.

**Scope and limitations.** Despite its effectiveness, CoT introduces extra computation (video segmentation and intermediate generations), increasing inference latency. Moreover, its performance depends on the base model's perception and generation capacity: stronger models yield better CoT-based reasoning.

Table 8: Holistic OCR vs. multilingual supervision during instruction tuning (LLaVA-One-Vision).

| Language | Phase 1 | Phase 2 | Phase 3 |
|---|---|---|---|
| English | 34.2 | 36.7 | 39.2 |
| Chinese | 22.3 | 26.4 | 29.1 |
| Korean | 12.1 | 13.5 | 16.0 |
| Japanese | 10.0 | 11.2 | 14.2 |
| German | 8.5 | 8.2 | 14.3 |

Table 9: Ablation of CoT strategies on Qwen2.5-VL-7B. HR, LR, TR, SR denote Holistic, Local, Temporal, and Spatial Reasoning respectively.

| Method | HR ↑ | LR ↑ | TR ↑ | SR ↑ |
|---|---|---|---|---|
| Baseline | 36.0 | 26.5 | 35.4 | 35.2 |
| Partial-Time CoT | 36.2 | 26.3 | 35.4 | 35.3 |
| Partial-Task CoT (VS only) | 37.3 | 27.1 | 35.9 | 37.2 |
| Partial-Task CoT (VTS only) | 38.5 | 27.5 | 36.8 | 38.4 |
| **Full CoT (ours)** | **40.5** | **28.7** | **37.2** | **40.9** |

# H    COLLECTING DETAILS OF VIDTEXT

This section outlines the procedures for sourcing, filtering, and analyzing the video content in VID-TEXT.

**Sources.**    To ensure a broad coverage of video scenarios and textual styles, VIDTEXT integrates data from six public datasets:

- **BOVText** (Wu et al., 2021) — Multi-scene videos suitable for holistic OCR tasks.
- **RoadText-1K** (Reddy et al., 2020) — Dense road-text detection in driving scenarios.
- **DSText** (Wu et al., 2023) — Subtitles from indoor instructional videos.
- **M4-ViteVQA** (Zhao et al., 2022) — Clip- and instance-level multimodal QA videos.
- **Video-MME/MLVU** (Fu et al., 2024; Zhou et al., 2024) — Long-form videos with strong temporal reasoning demands.

**YouTube supplementation.**    To supplement long-form data, we collect additional videos from YouTube, focusing on the following categories:

- **Sports highlights:** NBA, FIFA World Cup, and related competitions.
- **Gaming commentary:** live streams and post-game analysis.
- **TV shows and variety entertainment.**

**Retrieval and filtering criteria.**    Candidate videos were retrieved using targeted keyword queries such as *"match subtitles"*, *"game commentary"*, and *"captioned recap"*. We applied the following filtering rules:

- **Minimum duration:** ≥3 minutes for YouTube, >30 minutes for Video-MME.
- **Scene-text richness:** We use the latest detector **Gomatching** (He et al., 2024) to calculate the proportion of frames containing text.
- **Density thresholds:** Videos must meet a minimum ratio of text-bearing frames: **20%** for YouTube videos and **10%** for Video-MME.

**Metadata statistics.**    We also collect metadata such as video length, resolution, and frame rate to ensure coverage diversity across temporal and visual characteristics.

**Instance Annotation Guidelines**

**Step 1: Text Detection (Bounding Box)**
1.Draw a bounding box around 3-5 visible text instance in each video.
2.Annotate entire text lines, not individual words or characters.
3.If the same text appears across multiple frames, assign it the same Track ID using the tracking tool provided.

**Step 2: Text Classification**
Each bounding box must be assigned one of the following text categories:
Clear Text:clearly visible text.
Illegible: text that is unreadable due to blur, occlusion, or low resolution.

**Step 3: Text Transcription**
All Text instances require transcription.
For tracked text across multiple frames, you only need to transcribe it once—the tool will propagate it across the track automatically

**Special Handling: Blur or Occlusion**
If a text instance becomes blurred or occluded:
If the blur/occlusion lasts 3 frames or fewer, continue the original track.
If it lasts more than 3 frames, end the current track and create a new one labeled as Illegible.
If a text transitions from unreadable to readable (or vice versa), create a new track with the updated label

Figure 8: Instance-level annotation guidelines.

# I DETAILS OF ANNOTATION

## I.1 INSTANCE ANNOTATION

Each video underwent a two-stage text annotation process. In the first stage, annotators drew tight bounding boxes around visible text lines and assigned each to a category: *ClearText* or *Illegible*. A tracking tool automatically propagated bounding boxes across frames using consistent Track IDs. More details are shown in Fig. 8.

## I.2 CLIP-LEVEL ANNOTATION

Videos shorter than 1 minute were split into 5-second clips; longer ones into 20-second clips. For each clip, annotators recorded all visible, legible text and its temporal span. Repeated instances within a clip were marked only once. Illegible or heavily blurred texts were ignored. More details are shown in Fig. 9.

## I.3 VIDEO-LEVEL TEXT COLLECTION

A separate annotation team reviewed the OCR predictions from our model. Annotators removed hallucinated content and added missing instances. Chinese was annotated by full lines; other languages (e.g., English, German) were annotated by words. Each unique string was listed once in the final inventory. More details are shown in Fig. 9.

## I.4 HOLISTIC REASONING

Annotators watched the full video and consulted the video-level text inventory to write one multi-label question per video (see Fig. 10). Each question included seven options describing high-level semantics such as scene, role, topic, or sponsor.

## I.5 LOCAL REASONING

For every clip (as defined in D.2), annotators created one four-option multiple-choice question requiring reasoning between localized text and visual context (e.g., subtitle or character behavior). The

**Clip & Video-Level Annotation Guidelines**

**Clip Level:**

**Video Segmentation**

1.Divide the video into consecutive temporal segments：If the video is shorter than 1 minute:divide it into clips of **5 seconds** each.Else; divide it into clips of **20 seconds** each.

2.Each segment should have a clear start_time and end_time.

**Text Identification**

1.For each clip, annotate all readable text instances that appear within the clip's time span.

2.Ignore illegible, blurred, or heavily occluded text.

3.If the same text appears multiple times within the clip, annotate it only once.

**Video Level:**

**Global Text Collection**

1.Watch through the full video and record all clearly visible and legible text content.

2.You will be provided with a preliminary list of detected texts (from an automatic text detection model). In this case, carefully review and correct the list by adding missing texts and removing false positives to ensure accuracy.

3.Each unique text instance should be annotated only once (no need to mark repetitions).

**Language-Based Annotation Rules**

1.For Chinese text: annotate by complete text lines (e.g., subtitle or sign line).

2.For Non-Chinese languages (e.g., English, German): annotate by individual words.

3.For mixed-language cases, follow the dominant language rule and note exceptions when needed.

Figure 9: Clip- and video-level annotation guidelines.

**Annotation Guidelines for Holistic Reasoning**

**goal:**: Given the overall textual and visual content throughout the video—including information across multiple time segments—annotate a global question that requires semantic reasoning across time and space.

**Input**

You will be given the full video and its OCR transcription.

Your goal is to:Observe the entire video, noting important text and visual elements across different timepoints.Identify high-level topics, roles, actions, or patterns that emerge over time.

Create a multi-option question that tests understanding of the video's overall narrative or semantic structure, including content distributed across time.

Select 3 correct options from a set of 7 plausible answers.

**CoT Expectation:**

You should simulate how a model would connect multiple distributed cues,

such as:"The subtitle shows the name + stage text shows show name + outfit = talent show"

"Multiple timepoints include branding (e.g., sponsor, stage banner) → context clue"

"Introduction + mid-performance + audience shot = global understanding of scene"

**GOOD EXAMPLE:**

Question：Based on the video text and description, which of the following statements accurately describe the scene and content of the video?

"A": "The young performer is identified as a 12-year-old talent from a rural background.",

"B": "The show being referenced is \"中国达人秀\" (China's Got Talent).",

"F": "The show features a challenge round sponsored by \"海飞丝\""

Figure 10: Holistic reasoning annotation guidelines.

question must require multimodal reasoning and not be solvable using text or image alone. More details are shown in Fig. 11.

I.6    TEMPORAL CAUSAL REASONING

Given a reference text (e.g., scoreboard or subtitle), annotators identified the timestamp of its appearance, observed the following 3–30 seconds, and formulated a causal reasoning question. The answer was a single factual sentence describing the resulting action. Each QA pair was anchored to the cue's timestamp. More details are shown in Fig. 12.

**Annotation Guidelines for Local Reasoning**

**goal:**: Within a specific time segment of the video, reason over the text and visual context to answer a multimodal question grounded in localized semantics.

**Input**

You are given a **specific video segment** along with:
• Detected OCR text within the segment
• The corresponding video frames

Your task is to:

Understand the meaning and context of the visible text in the clip. Interpret surrounding visual content (e.g., characters, objects, layout)

Construct a multiple-choice question that tests the model's semantic understanding and reasoning ability

Provide 4 candidate options and select the correct answer

**CoT Expectation:**

Ask: what does the text cause / reflect / imply?

Simulate the model making the connection:

"If the subtitle says 'stay still', and the character hides behind a wall → he's afraid / threatened"

**GOOD EXAMPLE:**

Q: "In the clip, the text 'Final Round' is shown. What does it suggest about the competition?"
A: "The winner will be decided in this match."
Q: "When the subtitle says 'Don't move', what is the person doing?"
A: "They are hiding quietly behind the shelf."

Figure 11: Local reasoning annotation guidelines.

**Annotation Guidelines for Temporal Causal Reasoning**

**goal:** Track a specific text instance in the video, analyze the sequence of related events, and annotate a question–answer pair that reflects their causal relationships.

**Input**

1. Locate the reference text
• Find the timestamp where the given text appears clearly (e.g., scoreboard, sign, subtitle).
• Pause at that moment and record the text content and timestamp.
2. Observe what happens next
• Watch the following 3–30 seconds of the video.
• Identify any actions, changes, or reactions that may be caused by or related to the text.
3. Write the QA pair
Question: Frame a question that highlights the relationship (e.g., "what happened after…" / "how did the player respond to…").
Answer: Describe the actual action concisely and factually.

**CoT Expectation:**

you should consider the temporal progression: what happened after the text appeared, and why it might be related.
Example: a low score triggers a coach's timeout; a red light prompts braking.

**GOOD EXAMPLE:**

"question": "At a score of 105:83, what move did James make to score?",
 "answer": "He stole the ball and dunked it."
"question": "At the beginning of the game when the score was 0:0, how did the Warriors player score while being defended by Player 1?",
 "answer": "By scoring a three-point shot",

Figure 12: Temporal causal reasoning annotation guidelines.

## I.7 SPATIAL REASONING

As shown in Fig. 13, at a given timestamp, annotators located a reference text or entity and constructed a question requiring reasoning over its spatial relation to nearby visual elements (e.g., direction, proximity, interaction).

**Quality control.** All annotations underwent double review. Each item was cross-validated by a second annotator, and disagreements were resolved by expert adjudication. On a random sample of 200 items, we achieved an average inter-annotator agreement of **0.81** (Cohen's $\kappa$), indicating high reliability.

**Annotation Guidelines for Spatial Reasoning**

**goal:** At a specific timestamp, infer the spatial relationship between a text instance (or person) and surrounding visual elements—such as direction, relative position, or interaction.

**Input**

1. Locate the reference text
• Find the timestamp where the given text appears clearly (e.g., scoreboard, sign, subtitle).
• Pause at that moment and record the text content and timestamp.
2. Observe what happens next
• Watch the following 3–30 seconds of the video.
• Analyze the scene: what object or person is near, behind, or interacting?
3. Write the QA pair
Compose a multiple-choice or open-form reasoning question and answer

**CoT Expectation:**
you should consider Reason about spatial layout: who is positioned where, and what action is implied.
Use directional and functional cues: "behind", "to the right", "blocking", "following".

**GOOD EXAMPLE:**
"question": "When the score was 31:18 and 2:09 remained in the game, where was Player 8 located when attempting the three-point shot?",
"answer": "Bottom-middle of the image, right 45-degree three-point position"
"question": "With the score 0:0, who is the player defending Timber-wolves' Player 5 (white jersey)?",
"answer": "Player 31"

Figure 13: Spatial reasoning annotation guidelines.

## J    DETAILS OF EXPERIMENTAL SETTINGS

### J.1    MODEL CONFIGURATION

We outline the primary baselines evaluated on VIDTEXT. To ensure fair comparison across both open- and closed-source models, we explicitly standardize frame sampling and spatial resolution for each baseline, as summarized in Tab. 10.

For proprietary models such as GPT-4o, Gemini 1.5 (Pro and Flash), and GPT-4-Turbo, we follow their official or API-supported settings. `GPT-4o` supports up to ∼500 image inputs, for which we adopt a uniform sampling rate of 0.5 fps with an input resolution of $512 \times 512$ to accommodate most of our videos. `GPT-4-Turbo` is restricted to 16 frames, uniformly sampled across the video, and resized to the same resolution.

For open-source models, we align each configuration with their original public implementations. `VideoChat-Flash`, `Qwen2-VL (7B)`, and all `Qwen2.5-VL` variants (3B/7B/72B) operate under a 1 fps sampling strategy, with a maximum of 768 frames per video. Models supporting extended temporal contexts—such as `VideoLLaMA 3`, `InternVL 2.5`, and `LLaVA-OV`—are provided with 64 uniformly sampled frames, resized to $336 \times 336$. `ShareGPT4Video` also uses 64 frames, but with a reduced spatial resolution of $224 \times 224$. `LongVU` and `LongVA` are evaluated with sparse and extended frame settings. `LongVU` uses 1 fps sampling, while `LongVA` accepts up to 128 uniformly distributed frames. `MiniCPM-V2.6` applies a fixed 64-frame sliding window, following its official implementation.

### J.2    HUMAN PERFORMANCE STUDY

To assess the upper-bound of performance on VIDTEXT, we conducted a controlled human evaluation across all tasks in our benchmark. Three annotators with experience in video analysis and text recognition were recruited to answer a representative subset of questions spanning all eight task types. Each participant was given access to the full video content and instructed to answer using their best judgment, without time constraints. The average human accuracy across all tasks reaches **89.5%**, substantially outperforming all evaluated models. In particular, humans demonstrated near-perfect scores in holistic and local OCR, reasoning, and spatial understanding tasks, highlighting the gap between human-level comprehension and the capabilities of current multimodal large models. These results serve as a reference ceiling for future model development and underline the complexity

Table 10: Frame–sampling and input-resolution settings for baselines.

| Model | Size | Sampling | Resolution |
|---|---|---|---|
| *Proprietary MLLMs* | | | |
| GPT-4-Turbo | – | 16 frames | $512^2$ |
| Gemini 1.5 Flash | – | 1 fps | $512^2$ |
| GPT-4o | – | 0.5 fps | $512^2$ |
| Gemini 1.5 Pro | – | 1 fps | $512^2$ |
| *Open-source MLLMs* | | | |
| LongVU | 3 B | 1 fps | $448^2$ |
| Qwen2.5-VL | 3 B | 1 fps | $448^2$ |
| Video-XL-Pro | 7 B | 1 fps | $448^2$ |
| LongVA | 7 B | 128 frames | – |
| MiniCPM-V2.6 | 7 B | 64 frames | $448^2$ |
| VideoChat-Flash | 7 B | 1 fps | $448^2$ |
| Qwen2-VL | 7 B | 1 fps | $448^2$ |
| Qwen2.5-VL | 7 B | 1 fps | $448^2$ |
| VideoLLaMA 3 | 7 B | 64 frames | $336^2$ |
| ShareGPT4Video | 8 B | 64 frames | $224^2$ |
| Oryx-1.5 | 32 B | 64 frames | $336^2$ |
| LLaVA-OV | 72 B | 64 frames | $336^2$ |
| Qwen2.5-VL | 72 B | 1 fps | $448^2$ |
| InternVL 2.5 | 78 B | 64 frames | $336^2$ |

and nuance of the video-text understanding challenges posed by VIDTEXT. More details are shown in Tab. 2.

## J.3 EXPERIMENT ENVIRONMENT

All experiments are conducted on a server equipped with 4×NVIDIA A100 GPUs (80GB each). Model inference and evaluation are implemented in PyTorch with mixed-precision support.

## K MODEL PROMPTS

Fig. 14 shows the prompt template used to obtain detailed frame-level captions from the Aria model. The prompt includes instructions to describe the scene, detect visible text, summarize actions, and relate them spatially and semantically. Tab. 11 lists the standardized prompt templates used for each task in VIDTEXT.

## L MORE VISUALIZATION RESULTS

We present additional visualizations of our VIDTEXT annotation examples in Fig. 15, Fig. 16, and Fig. 17.

## M ETHICS AND RESPONSIBLE DATASET USE

### M.1 CONSENT AND COMPENSATION FOR HUMAN ANNOTATORS

Our human evaluation involved **three graduate research assistants**. Prior to participation, all annotators were provided with an **Information Sheet** that clearly explained: (i) the **research purpose**; (ii) the nature of the **data to be viewed**; (iii) estimated **workload and duration**; (iv) **voluntary participation** with the right to withdraw at any time; and (v) assurance that **no personal information** would be collected and that results would be reported in **aggregate form**. All annotators **gave informed consent** by signing the document and were compensated at **\$25/hour**, following our institution's standard rate for annotation tasks. No demographic or identity-related information was

---

**Aria Caption Generation Prompt**

You are given images sampled from a video. Please imagine yourself in the scene and describe in detail what you see from your viewpoint. Your description should focus on the following aspects:

1. What is the overall scene or environment?
2. What visible objects or people are present?
3. Are there any texts (e.g., signs, labels, instructions)? If yes, what do they say?
4. What activities or actions are happening in the scene?
5. Are there any meaningful relationships between the scene texts and the objects, people, or actions around them?

Please write the description in a natural and informative way, as if explaining what you are currently seeing. Avoid mentioning "image" or "frame", and do not speculate beyond what is visible.

Output format:
- Scene description: [...]
- Visible texts: [...]
- Human and object activities: [...]
- Spatial or semantic relationships (if any): [...]

---

Figure 14: Prompt template used for Aria to generate frame-level captions.

collected, and no audio/video recordings were made. All annotation files are stored on **encrypted drives** accessible only to the author team.

Based on international guidelines (**US 45 CFR 46**, **EU GDPR**, and our institution's ethics policy), this activity is classified as **Not Human-Subjects Research (Not-HSR)** or **Exempt Category #4 (publicly available / anonymized data)**. We completed an **internal ethics self-assessment**, and will include the **signed consent forms** and self-review documentation in the supplementary materials.

## M.2 Privacy and Copyright Compliance for YouTube Videos

Regarding the **76 YouTube videos** (sports and esports content) used in our dataset: (i) all videos are sourced from **publicly accessible broadcasts** where faces are generally indistinct; (ii) for any identifiable individuals, we apply **automatic face blurring** and **crop out channel identifiers**; (iii) derived data are **downsampled in resolution** and **watermarks removed**; (iv) the dataset is released strictly for **non-commercial academic research** under the **CC-BY-NC-SA 4.0** license.

We adopt a **takedown policy**: a dedicated **contact email** will be provided on the dataset homepage and GitHub; upon request from content creators or copyright holders, we will **remove the relevant video within 48 hours**; in cases of full takedown, we will retain only **sparse sampled frames** or **annotations/metadata**, following the practice of datasets like **MovieNet** (Huang et al., 2020). A detailed **Usage and Takedown Policy** is included in the supplementary materials to ensure **privacy protection**, **responsible use**, and **copyright compliance**.

Table 11: Prompt templates used for VIDTEXT tasks.

| Task | Prompt template |
|---|---|
| **Holistic OCR** | "Recognize all visual texts in the video.
If the text is not in English, do not provide an English translation.
Do not include any descriptions, narrative, or context.
Output only the extracted text lines, each on a new line." |
| **Holistic Reasoning** | "Watch the video carefully and select the correct three answers.
Question: {question}
Options: {options}
Please output your answer in the format: `Correct Answers: A, B, C`" |
| **Local OCR** | "Watch the video and answer the following question based on its content.
Question: {question}
Please output only the texts that appear in the specified time interval as a JSON array of strings,
with each element representing one piece of text. Do not include any additional description or translation." |
| **Local Reasoning** | "Watch the video and answer the following multiple-choice question based on its content.
Question: {question}
Options:
Option A: {text}
Option B: {text}
...
Please select the correct option." |
| **Text Localization** | "Watch the video and answer the following question based on its content.
Please provide the time interval (in seconds, precise to 0.1s) during which the text appears in the video.
Output your answer in JSON format with keys 'start' and 'end'. For example: `{"start": 0.0, "end": 30.0}`.
Do not include any extra commentary." |
| **Temporal Causal Reasoning** | "Watch the video and answer the following multiple-choice question based on its content.
Question: {question}
Options:
Option A: {text}
Option B: {text}
...
Please select the correct option." |
| **Text Tracking** | *(Same prompt as Spatial Reasoning)* |
| **Spatial Reasoning** | "Watch the video and answer the following multiple-choice question based on its content.
Question: {question}
Options:
Option A: {text}
Option B: {text}
...
Please select the correct option." |

Figure 15: (Top) more examples of HolisticOCR. (Middle) more examples of HolisticReasoning. (Bottom) more examples of LocalOCR.

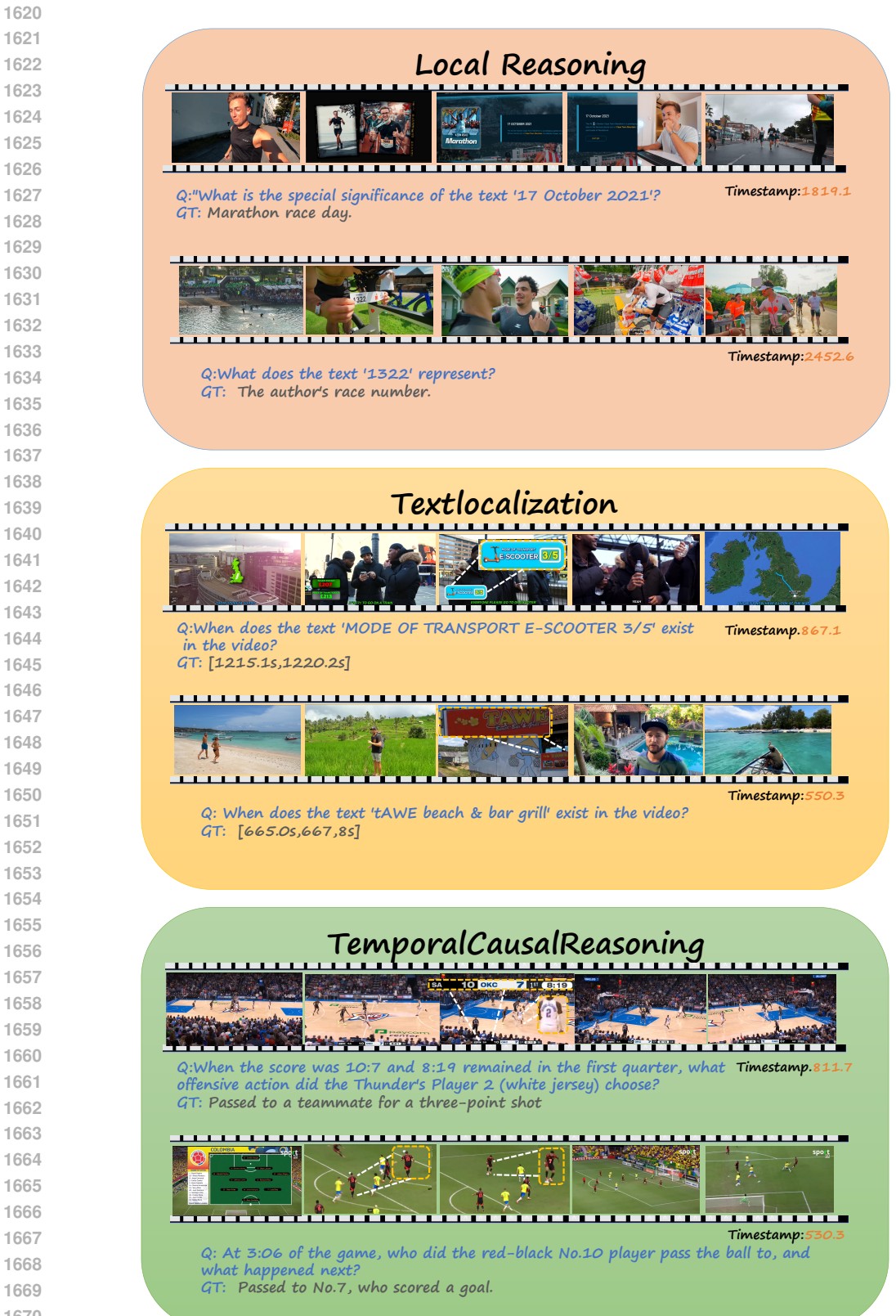

Figure 16: (Top) more examples of LocalReasoning. (Middle) more examples of TextLocalization. (Bottom) more examples of TemporalCausalReasoning.

Figure 17: (Top) more examples of TextTracking. (B ottom) more examples of SpatialReasoning.

