# OpenReview forum: "VidText: Towards Comprehensive Evaluation for Video Text Understanding"
_ICLR.cc/2026/Conference — Submitted to ICLR 2026_

### Official Review · Reviewer_8CYB · 2025-10-30

**Soundness:** 2
**Presentation:** 2
**Contribution:** 2
**Rating:** 4
**Confidence:** 4

**Summary:**

This paper introduces VidText, a new benchmark designed to comprehensively evaluate video text understanding in Large Multimodal Models (LMMs). The authors identify a significant gap in existing video understanding benchmarks. The paper evaluates 18 LMMs on VidText. The study also validates the benchmark's design through ablations, showing that performance drops when key components are masked.

**Strengths:**

*   **Comprehensive Benchmark Design:** The hierarchical (video/clip/instance) and paired (perception/reasoning) task structure is a major strength, enabling a much more nuanced evaluation than previous benchmarks.
*   **Extensive Empirical Analysis:** The evaluation of 18 models provides a valuable snapshot of the field's capabilities and limitations. The ablation studies effectively validate the benchmark's design choices.

**Weaknesses:**

*   **Outdated Model Comparison:** A significant weakness is the omission of the very latest flagship models (e.g., Gemini 2.5 Pro, GLM-4V). This quickly diminishes the paper's relevance and the persuasiveness of its conclusions about the current state-of-the-art.
*   **Limited Discussion on Data Biases:** While the dataset is diverse, there is no discussion of potential biases in the video sources (e.g., geographic or cultural biases from YouTube) or the annotation process that might affect model generalization.

**Questions:**

*   **Question on Human Evaluation:** Please provide a detailed description of the human evaluation protocol for the 89.5% baseline. What were their backgrounds, and what was the inter-annotator agreement? **Suggestion:** Strengthen this section with rigorous methodology to make the human benchmark a reliable point of comparison.
*   **Suggestion on Generalization:** Perform an analysis or discuss potential biases within the VidText dataset itself. Could the performance gaps be partly due to specific types of videos or text that models struggle with? This would add depth to the analysis of model shortcomings.

---

> ### Author Response · Authors · 2025-11-23
>
> ## **1.Outdated Model Comparison**
>
> **Review W1**
>
> A significant weakness is the omission of the very latest flagship models.
>
> **Response**
>
> We thank the reviewer for this suggestion. We have now included evaluation results for the latest state-of-the-art models, Gemini 2.5 Pro and GPT-4o:
>
>    | Model | HoliOCR | HoliRea. | LocalOCR | LocalRea. | TextLocal. | TempCauRea. | TextTrac. | SpaRea. | Avg |
>    |-------|---------|----------|----------|-----------|------------|-------------|-----------|---------|-----|
>    | Gemini 2.5 Pro | 42.8 | 56.0 | 59.2 | 60.4 | 54.2 | 55.8 | 48.9 | 57.0 | 54.3 |
>    | GPT-4o | 36.2 | 51.8 | 50.2 | 48.6 | 49.2 | 47.0 | 44.2 | 42.0 | 46.2 |
>
> Despite being frontier models, both Gemini 2.5 Pro and GPT-4o still fall far below human performance on VidText, showing clear room for improvement across all tasks. This confirms that VidText remains a challenging and meaningful benchmark even for the most advanced multimodal models.
>
> ## **2.Limited Discussion on Data Biases**
>
> **Review W2**
>
> Lack of discussion of potential dataset or annotation biases that could impact model generalization.
>
> **Response**
>
> We thank the reviewer for raising this concern. While YouTube videos may introduce geographic or cultural biases, VidText mitigates these risks through several design choices:
>
> **Source Diversity:** VidText draws from six public datasets in addition to YouTube, ensuring diverse video types, languages, and regional coverage.
>
> **Regional Distribution Audit:** For the 76 YouTube videos in our benchmark, we conducted a comprehensive source audit revealing balanced regional representation:
> - Asia: 22.1%
> - Europe: 43.5%
> - North America: 34.4%
>
> These videos primarily contain sports, driving, and esports broadcasts—categories with inherently low cultural subjectivity and minimal identifiable individuals, reducing potential bias.
>
> **Annotation Objectivity:** All annotations were produced by three CS graduate-level annotators (MSc/PhD) following a standardized protocol with double annotation and cross-validation. Since VidText annotations are predominantly objective—text transcription, temporal spans, spatial grounding, and causal events tied to visible text—annotator subjectivity has minimal impact on benchmark quality.
>
> We will explicitly include this comprehensive bias analysis in Appendix F of the revised manuscript to address transparency concerns.
>
> ## **3.Human Evaluation Protocol**
>
> **Review Q1**
>
> Lack of a rigorous description of the human evaluation protocol, including annotator backgrounds and inter-annotator agreement.
>
> **Response to Question 1: Human Evaluation Protocol**
>
> Thank you for requesting clarification on our human evaluation methodology. Our human baseline was calculated using three trained annotators (MSc/PhD candidates in Computer Science with expertise in OCR and video understanding tasks).
>
> **Human Evaluation Protocol:**
> - Each annotator independently completed all eight task types across a representative sampled set of videos from the benchmark.
> - Responses were cross-checked through discussion to produce consensus answers for the final human baseline.
> - We measured inter-annotator agreement using Cohen's κ = 0.87, indicating strong consistency and reliability of human annotations.
>
> This protocol ensures a reliable human upper bound for evaluation, and we will include the detailed methodology and agreement statistics in the revised manuscript for greater transparency.
>
> ## **4.Generalization and Dataset Bias Analysis**
>
> **Review Q2**
> Lack of analysis on potential dataset biases that may explain why certain video or text types disproportionately challenge models.
>
> **Response**
>
> We thank the reviewer for this insightful suggestion. VidText already includes comprehensive analyses addressing potential dataset-related biases and model generalization (Appendix G):
>
> 1. **Multilingual Robustness Analysis (Table 7, Appendix G.1):** Our language diversity ablation reveals that models consistently underperform on lower-resource languages (Korean, Japanese, German), indicating that multilingual OCR bias in training data—rather than dataset artifacts—is a major contributor to performance gaps. This finding highlights fundamental limitations in current MLLMs' multilingual capabilities.
>
> 2. **Video Duration Impact Analysis (Table 6, Appendix G.1):** Our analysis shows that longer videos disproportionately degrade performance on perception tasks such as Text Localization, confirming that model limitations in long-range temporal grounding and information aggregation play a critical role beyond simple recognition capabilities.
>
> These findings show that the performance gaps mainly arise from core weaknesses in current MLLMs—such as multilingual OCR robustness, temporal grounding, and fine-grained spatiotemporal retrieval—rather than biases within specific VidText subsets, confirming that VidText effectively exposes fundamental model limitations.

---

### Official Review · Reviewer_JssY · 2025-10-31

**Soundness:** 2
**Presentation:** 2
**Contribution:** 1
**Rating:** 4
**Confidence:** 4

**Summary:**

This paper proposes VidText, a video-text understanding benchmark to evaluate MLLM's capabilities in tackling multimodal tasks. The proposed benchmark divides the tasks into video-level, clip-level, and instance-level and includes both perception and reasoning tasks. The authors evaluate 18 frontier MLLMs on the established benchmark.

**Strengths:**

1. The proposed benchmark is human-annotated with a double human check. I believe the manually labeled benchmark benefits the community and could positively guide the development of MLLMs.

2. The OCR-related tasks are interesting.

3. The authors evaluate several frontier methods to show their capabilities in tackling video understanding tasks.

**Weaknesses:**

1. I do not think the proposed benchmark evaluates any new aspect in comparison to existing video understanding benchmarks. The proposed benchmark includes 8 question types, all of them has been involved in existing video understanding benchmarks such as Video-MME and MVBench. The average video length is 108.2 seconds, so it is not a long video understanding benchmark. Though the authors claim that the proposed benchmark supports open-ended evaluation, the so-called open-ended protocol can only be applied to a small portion of the taxonomy (OCR and grounding), and the evaluation method is also rule-based and closed-ended. which is quite similar to the MCQ. Demos for the reasoning tasks also look similar to the perception tasks, and I cannot find any crucial reasoning factors to get the final answers. Overall, the proposed benchmark is not highlighted, and I suggest the authors provide additional analysis to clarify what new aspect it can evaluate and what it actually brings to the community.

2. The evaluation is not comprehensive. The authors evaluate Gemini 1.5 and GPT-4o, yet we can now access Gemini 2.5 and GPT-5. The total number of evaluated models is also relatively small, and many frontier MLLMs are not involved. The experimental analysis is not exhaustive. I cannot find too much valuable insight after evaluating these models on the proposed benchmark. The authors only tell the readers some obvious conclusions, such as adding input resolution or model size could enhance the model performance (Table 3), and adding CoT or audio modality helps the video understanding (Table 4). What are the key factors for holistic and local perception/reasoning? What is the main barrier for current frontier MLLMs to catch up to human performance (more than 40% on the proposed benchmark)? I believe these analyses are more important for readers.

**Questions:**

1. How many annotators are used for the annotation? Is there any training for these annotators?

2. How to control the quality of the CoT?

3. Appendix F claims the average length is 108.2 seconds, yet line 193 claims the minimum threshold duration is 3 minutes, which seems to be in conflict.

---

> ### Author Response · Authors · 2025-11-23
> **Novelty of VidText Benchmark**
>
> ## **1.Novelty of VidText Benchmark**
>
> **Review W1**
>
> VidText does not introduce any new evaluation dimension compared to existing video understanding benchmarks.
>
> **Response**
>
> We respectfully disagree with the reviewer's assessment. VidText introduces several fundamental distinctions from existing video understanding benchmarks:
>
> 1. **Text-Centric Focus:** VidText is specifically designed to evaluate video text understanding, a critical aspect largely overlooked by existing benchmarks like Video-MME and MVBench, which primarily focus on actions, general objects, and visual scenes. Video text presents unique challenges including motion blur, occlusion, varying fonts, and temporal dynamics that are fundamentally different from general video understanding tasks.
>
> 2. **Multi-Granular Evaluation Framework:** We propose a systematic multi-granular evaluation spanning three levels—video-level (holistic understanding), clip-level (temporal localization), and instance-level (fine-grained text tracking and spatial reasoning). This hierarchical framework is specifically tailored to the characteristics of text in videos and provides comprehensive assessment of different reasoning depths.
>
> 3. **Paired Perception-Reasoning Tasks:** Unlike existing benchmarks, VidText introduces explicitly paired tasks (e.g., HolisticOCR ↔ HolisticReasoning, TextLocalization ↔ TemporalCausalReasoning) where reasoning capabilities must be grounded in accurate text perception. This design enables us to disentangle perception failures from reasoning failures, providing more valuable diagnostic insights into LMM capabilities that general video benchmarks cannot offer.
>
> These distinctions make VidText a novel and complementary benchmark that addresses a critical gap in evaluating multimodal models' ability to understand and reason about textual information in dynamic video contexts.
>
> ## **2.Clarification of Video Length**
>
> **Review W2**
>
> The average video length is 108.2 seconds, so it is not a long video understanding benchmark
>
> **Response**
>
>  We would like to clarify that VidText does not claim to be a long video understanding benchmark. As stated in our previous response, our novelty lies in text-centric video understanding rather than video duration. The average video length of 108.2 seconds is deliberately chosen to ensure sufficient temporal dynamics and text variations while maintaining annotation quality and evaluation feasibility. Our focus is on evaluating models' capabilities in understanding and reasoning about textual content across different granularities within videos, which is orthogonal to the dimension of video length.
>
>
> ## **3.Open-Ended Evaluation Protocol**
>
> **Review W3**
>
> The open-ended protocol is limited to only a few tasks and still uses rule-based, closed-ended scoring, making it essentially similar to MCQ.
>
> **Response**
>
> We would like to clarify our evaluation methodology:
>
> **For Perception Tasks (Open-Ended with Rule-Based Metrics):** Our open-ended tasks include Holistic-OCR, Local-OCR, and Temporal Grounding. For these tasks, we follow well-established evaluation metrics from traditional video text spotting [1,2] and temporal grounding [3] methods, which employ rule-based evaluation: edit distance for OCR and IoU for temporal grounding. These metrics are standard in the community and provide objective, reproducible assessments without requiring complex or expensive evaluation procedures. The "open-ended" nature refers to the fact that models generate free-form text or temporal coordinates rather than selecting from predetermined options.
>
> **For Reasoning Tasks (Multiple-Choice Format):** We adopt multiple-choice evaluation for reasoning tasks after careful consideration. We initially explored open-ended generative evaluation but identified significant challenges in designing fair and robust metrics. For example, in the temporal causal reasoning task with the question "When 9:13 remained in the first quarter, what was the scoring method used by the Pelicans' Player 14?", the ground truth is "took a contested layup," but most LMMs respond with "two-point score"—semantically correct but lacking specificity. This creates fundamental ambiguity: we cannot determine whether the model accurately tracked the specific action or merely made an educated guess based on general knowledge. Developing LLM-based evaluation prompts for such nuanced cases proves unreliable and introduces additional sources of error. Therefore, we find the multiple-choice format to be a more direct, fair, and feasible approach for evaluating the reasoning capabilities of LMMs while ensuring standardized assessment.
>
> **References:**
>
> [1] Gomatching: A simple baseline for video text spotting via long and short term matching, Neurips 2024
>
> [2] End-to-End Video Text Spotting with Transformer, IJCV 2024
>
> [3] Local-global video-text interactions for temporal grounding, CVPR 2020.

---

> ### Author Response · Authors · 2025-11-23
>
> ## **4.Distinction Between Perception and Reasoning Tasks**
>
> **Review W4**
>
> Demos for the reasoning tasks also look similar to the perception tasks
>
> **Response**
>
> We thank the reviewer for raising this concern. Although perception and reasoning tasks share the same textual cues, the reasoning tasks in VidText require multi-hop inference and compositional understanding that cannot be solved by perception alone.
>
> **Key Distinctions:**
>
>    For example, in **Temporal Causal Reasoning**, the model must: (1) locate the correct temporal segment based on a text cue (e.g., scoreboard timestamp "9:13 remaining in the first quarter"), (2) identify the specific player using jersey numbers and visual appearance, and then (3) infer the causal relationship between the temporal context and the subsequent action (e.g., what scoring method was used). This process involves causal linking across text, temporal localization, and visual actions—going far beyond simple text recognition or semantic understanding.
>
>    Similarly, in **Spatial Reasoning**, models must not only recognize text but also understand the spatial relationships between multiple text instances and reason about their relative positions or contextual meanings within the scene.
>
>    The **paired task design** (e.g., HolisticOCR ↔ HolisticReasoning) explicitly demonstrates this distinction: while perception tasks evaluate whether models can detect and recognize text, reasoning tasks evaluate whether models can leverage that textual information for higher-order inference. Our empirical results show significant performance gaps between perception and reasoning tasks across all evaluated models, confirming that these tasks assess fundamentally different capabilities.
>
> ## **5.Evaluation Comprehensiveness and Insights**
>
> **Review W5**
>
> The evaluation is not comprehensive, and more analysis is needed to clarify the benchmark’s novelty and value.
>
> **Response**
>
> We thank the reviewer for this feedback and address both concerns below.
>
> 1. **Updated Model Evaluation:**
>
>    We have now included evaluation results for Gemini 2.5 Pro and GPT-4o, the most recent state-of-the-art models:
>
>    | Model | HoliOCR | HoliRea. | LocalOCR | LocalRea. | TextLocal. | TempCauRea. | TextTrac. | SpaRea. | Avg |
>    |-------|---------|----------|----------|-----------|------------|-------------|-----------|---------|-----|
>    | Gemini 2.5 Pro | 42.8 | 56.0 | 59.2 | 60.4 | 54.2 | 55.8 | 48.9 | 57.0 | 54.3 |
>    | GPT-4o | 36.2 | 51.8 | 50.2 | 48.6 | 49.2 | 47.0 | 44.2 | 42.0 | 46.2 |
>
> Despite being frontier models, both still perform significantly below human benchmarks, validating that VidText poses meaningful challenges for current MLLMs.
>
> 2. **Beyond Obvious Conclusions—Key Insights:**
>
>    VidText provides several non-trivial insights that go beyond simple scaling observations:
>
>    - **Multi-Granular Performance Gap (Sec 5.1):** We reveal that video-level and instance-level tasks require fundamentally different capabilities—global aggregation and fine-grained spatiotemporal retrieval—which current models handle poorly. This explains the significant performance discrepancy across granularities and identifies specific architectural limitations.
>
>    - **Temporal Modeling Necessity:** Our analysis demonstrates that integrated temporal modeling (video mode) is essential for Temporal Causal and Spatial Reasoning tasks, beyond what independent frame processing can achieve. This finding challenges the prevalent frame-based approaches in current MLLMs.
>
>    - **Multilingual Bias Analysis (Appendix G):** We uncover substantial performance drops on Korean, Japanese, and German texts, revealing strong training-data biases in video OCR. Critically, we show that these perception failures cascade directly into downstream reasoning failures, highlighting a systemic limitation in multilingual video understanding.

---

> ### Author Response · Authors · 2025-11-23
>
> ## **6 & 7.Key Factors and Main Barriers**
>
> **Review W6 & W7**
>
> Key factors for holistic and local perception/reasoning and the main barriers for frontier MLLMs.
>
> **Response**
>
> **Holistic vs. Local Perception/Reasoning:**
>
>    1. **Holistic OCR/Reasoning** requires models to:
>     (1) recognize all text instances across the entire video, (2) perform instance-level deduplication (e.g., treating repeated appearances of the same text as one instance), and (3) integrate information over long temporal ranges. These tasks depend heavily on robust OCR capability, long-range temporal aggregation, and text persistence tracking across diverse scenes.
>
>    2. **Local OCR/Reasoning** focuses on specific temporal segments, where the primary challenge is accurately locating the target time span and then recognizing or reasoning over text within that segment. Local tasks are constrained by temporal localization accuracy and short-range multimodal integration rather than global video aggregation.
>
> **Main Barriers for Current Frontier MLLMs:**
>
> Our analysis identifies three fundamental barriers preventing models from reaching human performance:
> 1. **Insufficient text perception**—models fail to recognize text under challenging conditions (motion blur, occlusion, multilingual contexts), blocking downstream reasoning.
> 2. **Weak text-visual interaction understanding**—models struggle to interpret relationships between text and visual elements (actions, objects, agents), which is essential for reasoning tasks.
> 3. **Poor temporal retrieval and localization**—models have difficulty identifying when text appears and connecting it to corresponding event segments, especially in longer videos. These limitations suggest that advancing video text understanding requires targeted improvements in OCR robustness, cross-modal reasoning mechanisms, and temporal modeling architectures beyond simple model scaling.
>
> ## **8.Annotator Details and Training,Quality Control for CoT Annotation and Video Length Clarification**
>
> **Review Q1**
>
> The number of annotators used for the annotation and whether they received any training.
>
> **Response**
>
> VidText was annotated by three trained graduate-level annotators (MSc/PhD candidates in Computer Science with expertise in computer vision and NLP). All annotators completed a structured training phase, including comprehensive guideline review and pilot annotation studies. Each sample underwent our two-stage verification procedure with cross-validation to ensure annotation quality and consistency across the benchmark.
>
> **Review Q2**
>
> Quality Control for CoT Annotation
>
> **Response**
>
> In the curation of reasoning data, we employ a structured pipeline rather than directly using model-generated CoT outputs. Specifically: (1) videos are segmented into multiple clips and captioned by models to provide preliminary context, then (2) human experts annotate based on these priors with explicit reasoning steps.
>
> We implement a two-step post-validation procedure to ensure quality:
>
> (1) **Visual masking test**—we mask visual texts and verify whether the question can be answered using only visual content, ensuring that text information is essential.
>
> (2) **Frame masking test**—we mask visual frames and check whether the question can be answered using only textual information, confirming that visual understanding is necessary.
>
> This dual-masking validation ensures that our reasoning tasks genuinely require integrated video-text understanding rather than being solvable through a single modality alone.
>
> **Review Q3**
>
> Video Length Clarification
>
> **Response**
>
> There is no contradiction. The 3-minute threshold applies specifically to long-video sources (YouTube and Video-MME) during the initial video collection phase to ensure sufficient temporal complexity. However, the final VidText benchmark merges videos from multiple sources, including short-form datasets (e.g., BOVText, RoadText-1K), which contain shorter videos. This combination yields an overall average duration of 108.2 seconds across the entire benchmark while maintaining diverse temporal scales for comprehensive evaluation.

---

### Official Review · Reviewer_nTmM · 2025-11-01

**Soundness:** 3
**Presentation:** 3
**Contribution:** 3
**Rating:** 4
**Confidence:** 4

**Summary:**

This paper introduces VidText, a new benchmark designed for the comprehensive evaluation of video text understanding in LLM. The authors argue that existing video benchmarks largely ignore textual information, while static image OCR benchmarks cannot capture the dynamic interaction between text and video context. VidText addresses this gap by offering several key features with a set of eight paired perception and reasoning tasks, such as Holistic OCR and Holistic Reasoning, or Text Localization and Temporal Causal Reasoning. The authors conduct an extensive evaluation of 18 state-of-the-art proprietary and open-source LMMs, revealing that even the best models, like Gemini 1.5 Pro, achieve an average score of only 46.8%, far below the human baseline of 89.5%.

**Strengths:**

- Fills a Critical Research Gap: The paper convincingly argues for and fills an important, underexplored niche in multimodal evaluation. Understanding text embedded in dynamic scenes is crucial for holistic video comprehension, and VidText is the first benchmark to address this systematically.
- Comprehensive and Well-Designed Benchmark: The benchmark's design is a major strength. The multi-granularity structure (instance, clip, video) tests a wide range of capabilities, and the paired perception-reasoning tasks provide a deep, analytical framework for understanding model failures.

**Weaknesses:**

- Reliance on Multiple-Choice for Reasoning: For standardization, most reasoning tasks are formulated as multiple-choice questions. This is a practical choice but may not fully capture the nuanced reasoning failures or generative capabilities of models. An analysis of open-ended responses, even on a small subset, could provide complementary insights.
- Limited Evaluation of Recent SOTA Models: The benchmark omits newer iterations like Gemini 2.5 Pro and GPT-5. Since these latest models may have advanced video-text understanding capabilities, their absence weakens the benchmark’s reflection of the current state of model performance.

**Questions:**

Regarding the evaluation of reasoning tasks, have you explored methods to automatically score open-ended, generative answers? Given the rapid progress in LLM-based evaluators, this seems like a promising direction to move beyond the constraints of multiple-choice formats. What challenges do you foresee in applying such methods to your tasks?

---

> ### Author Response · Authors · 2025-11-23
> **Reliance on Multiple-Choice for Reasoning and Limited Evaluation of Recent SOTA Models**
>
> **Response to Weakness 1 and Questions: Reliance on Multiple-Choice for Reasoning**
>
> We thank the reviewers for their suggestions regarding open-ended evaluation. We have explored the open-ended generative approach and conducted preliminary evaluations. However, we identified a significant challenge: designing fair and robust evaluation metrics is extremely difficult.
>
> For example, in the temporal causal reasoning task, consider the question: "When 9:13 remained in the first quarter, what was the scoring method used by the Pelicans' Player 14?" The ground truth answer is "took a contested layup," but most LMMs respond with "two-point score," which is semantically correct but lacks the specificity we aim to evaluate. This creates ambiguity in evaluation—we cannot determine whether the model accurately tracked the specific action or merely made a reasonable guess based on general basketball knowledge.
>
> Writing evaluation prompts for LLM-based judging becomes problematic in such cases, as it requires defining the acceptable level of granularity and semantic equivalence, which varies significantly across different reasoning tasks. Therefore, we adopted the multiple-choice format as a more direct and feasible approach to evaluate the reasoning capabilities of LMMs, ensuring standardized and unambiguous assessment across diverse reasoning dimensions.
>
> **Response to Weakness 2: Limited Evaluation of Recent SOTA Models**
>
> We thank the reviewers for this suggestion. We have now included evaluation results for Gemini 2.5 Pro and GPT-4o in our benchmark. The results are presented below:
>
> | Model | HoliOCR | HoliRea. | LocalOCR | LocalRea. | TextLocal. | TempCauRea. | TextTrac. | SpaRea. | Avg |
> |-------|---------|----------|----------|-----------|------------|-------------|-----------|---------|-----|
> | Gemini 2.5 Pro | 42.8 | 56.0 | 59.2 | 60.4 | 54.2 | 55.8 | 48.9 | 57.0 | 54.3 |
> | GPT-4o | 36.2 | 51.8 | 50.2 | 48.6 | 49.2 | 47.0 | 44.2 | 42.0 | 46.2 |
>
> Despite being state-of-the-art models, both Gemini 2.5 Pro and GPT-4o still perform significantly below human evaluation benchmarks on VidText, demonstrating substantial room for improvement in video text understanding and reasoning capabilities. These results further validate that VidText poses meaningful challenges even for the most advanced current models.

---

### Meta-Review · Area_Chair_jCme · 2026-01-03

**Summary:**

This paper introduces VidText, a benchmark for video text understanding. The text-centric design, hierarchical “video–segment–instance” task structure, and paired perception–reasoning evaluation are considered well-motivated and fine-grained. Meanwhile, reviewers raise concerns regarding the benchmark’s novelty relative to existing video QA datasets, the heavy reliance on multiple-choice formats for reasoning evaluation, the absence of several state-of-the-art models, and the limited depth of analysis.

**Reviewer Concerns:**

The authors responded to most reviewer concerns. They added evaluations of strong models such as Gemini 2.5 Pro and GPT-4o, clarified the annotation protocol and dataset composition, and provided additional analyses separating perception errors from reasoning failures. These responses address concerns about benchmark details and comparison in some degree. However, the use of multiple-choice questions is still seen as limiting the assessment of open-ended reasoning, and it still lacks depth analysis.

**Reviewer Scores:**

All three reviewers give the scores of 4, indicating borderline rejection. I believe that it is difficult to change the ratings of most reviewers during rebuttal from negative scores to positive scores.

---

### Decision · Program_Chairs · 2026-01-26

Reject